# FOLLOW-THE-PERTURBED-LEADER FOR ADVERSARIAL BANDITS: HEAVY TAILS, ROBUSTNESS, AND PRIVACY

## ABSTRACT

We study adversarial bandit problems with potentially heavy-tailed losses. Unlike standard settings with non-negative and bounded losses, managing negative and unbounded losses introduces a unique challenge in controlling the "stability" of the algorithm and hence the regret. To tackle this challenge, we propose a Follow-the-Perturbed-Leader (FTPL) based algorithm. Notably, our method achieves (near-)optimal worst-case regret, eliminating the need for an undesired assumption inherent in the Follow-the-Regularized-Leader (FTRL) based approach proposed in the prior work. Thanks to this distinctive advantage, our algorithmic framework finds novel applications in two important scenarios with unbounded heavy-tailed losses. For adversarial bandits with heavy-tailed losses and Huber contamination, which we call the robust setting, our algorithm is the first to match the lower bound (up to a $\mathrm{polylog}(K)$ factor, where $K$ is the number of actions). In the private setting, where true losses are in a bounded range (e.g., $[0, 1]$) but with additional Local Differential Privacy (LDP) guarantees, our algorithm achieves an improvement of a $\mathrm{polylog}(T)$ factor in the regret bound compared to the best-known results, where $T$ is the total number of rounds. Furthermore, when compared to state-of-the-art FTRL-based algorithms, our FTPL-based algorithm has a more streamlined design. It eliminates the need for additional explicit exploration and solely maintains the absolute value of loss estimates below a predetermined threshold.

## 1 INTRODUCTION

Adversarial (or non-stochastic) Multi-armed Bandits (MAB) is a classic Online Learning problem with rich literature (Auer et al., 2002; Audibert & Bubeck, 2009). Adversarial MAB is formulated as a $T$-round repeated game between a learner and an adversary. In each round, the learner chooses (or plays) one of the $K$ actions, whose losses are determined by the adversary. The learner suffers a loss from the chosen action and observes it, while the losses of all the other actions remain unrevealed. The learner's goal is to minimize the *regret* (see Eq. (1) for the formal definition), which quantifies how well the learner performs compared to always choosing the best fixed action in hindsight.

While most works in this topic study the case where all the losses lie in a bounded range (most commonly, in $[0, 1]$), handling potentially unbounded losses is still a key challenge. Taming heavy-tailed losses, a special case of unbounded losses, has gained lots of research interest since real-world data often exhibits a heavy-tailed nature (Hamza & Krim, 2001; Rachev, 2003; Hull, 2012).

In adversarial bandits, the main challenge from heavy-tailed losses comes largely from *negative losses*, which are known to cause *"extremely-negative" loss estimates* (negative with large absolute value) (Dai et al., 2023). Such loss estimates might break the "stability" of a (randomized) algorithm. Here, stability roughly means that the (conditional) probability distributions over the actions from which the decision is generated should not change significantly over any two consecutive rounds, which is an essential property that ensures desirable regret guarantees. To ensure stability, Huang et al. (2022) proposes a Follow-the-Regularized-Leader (FTRL) based algorithm, which achieves the minimax-optimal regret, but requires an additional assumption of "truncated non-negativity" (see Assumption 2). This assumption is undesired since not only does it restrict the heavy-tailed distributions to a subclass, but it also prevents their algorithm from finding wider applications, e.g., our robust setting. On the other hand, while Follow-the-Perturbed-Leader (FTPL) has not been investigated in the heavy-tailed setup, a natural question is: Can one design a (nearly) optimal FTPL-based algorithm that bypasses the issues in FTRL-based prior work and finds broader applications?

**Our contributions.** Leveraging a key observation on an existing stability result of FTPL, we propose a learning algorithm under the FTPL framework, which allows us to still achieve the (nearly) minimax-optimal regret, and more importantly, bypass issues due to negative losses including the strong assumption required in Huang et al. (2022) (see details in Section 5). Furthermore, with the capability of handling heavy-tailed losses, our algorithm can further be applied to obtain near-optimal or improved regret guarantees in two important applications even with a simpler design (i.e., controlling the scale of loss estimates via "skipping"). In contrast, existing FTRL-based counterparts need more complicated design/analysis in addition to ours, or do not yield any non-trivial guarantees. The first application is adversarial bandits with heavy-tailed losses and Huber contamination (Huber, 1964), where in each round, with some constant probability $\beta > 0$, the feedback can be "contaminated", i.e., generated from an unknown and arbitrarily "bad" distribution rather than the original "clean" heavy-tailed distribution. The second one is adversarial bandits with bounded losses and Local Differential Privacy (LDP). We summarize our main contributions as follows:

**(i)** In the setup of heavy-tailed adversarial bandits where the losses' $\alpha$-th moment is bounded by $\sigma^\alpha$ for some constants $\alpha \in (1, 2]$ and $\sigma > 0$, we propose an FTPL-based algorithm and prove a (nearly) minimax-optimal regret bound of $O(\sigma K^{1-1/\alpha} T^{1/\alpha} \sqrt{\ln K})$,[1] which matches the lower bound (up to a $\sqrt{\ln K}$ factor) in Bubeck et al. (2013). More importantly, our key observation on the stability of FTPL (different from that of FTRL) allows us to naturally handle negative losses, without those issues under FTRL, e.g., additional assumptions, sophisticated analyses, and worse regret bounds, which we believe may find broader interests.

**(ii)** In the robust setting, i.e., heavy-tailed adversarial losses with Huber contamination, we show that our algorithm achieves a regret upper bound of $\widetilde{O}(\sigma K^{1-1/\alpha} T^{1/\alpha} + \sigma T \beta^{1-1/\alpha})$, which offers a first non-trivial bound in the adversarial setting and is near-optimal (see Section 8).

**(iii)** In the private setting, i.e., bounded losses (in $[0, 1]$) with LDP guarantees, our algorithm achieves a regret of $O(\sqrt{KT \ln K}/\varepsilon)$, where $\varepsilon \in (0, 1]$ is the privacy budget, which improves upon that of the FTRL-based algorithms in Agarwal & Singh (2017) and Tossou & Dimitrakakis (2017) by a factor of $\sqrt{\ln T}$ and $\ln T$, respectively (see Section 9).

## 2 RELATED WORK

Due to space limitations, we only discuss the most relevant lines of research here. See Appendix A for an elaborated discussion on related work.

**Follow-the-Perturbed-Leader.** FTPL is a classic framework, where the stability is ensured by randomly-perturbed cumulative losses. However, one hurdle of applying FTPL is that its analysis "lacks a generic framework and relies heavily on mathematical tricks" (Abernethy et al., 2016). Therefore, the (best-known) regret guarantee of FTPL is typically worse than that of FTRL, or even missing. While theoretical guarantees have been established from bandits to Markov Decision Processes (MDPs) (Neu & Bartók, 2016; Dai et al., 2022; Honda et al., 2023), all these works study the standard bounded losses case, and FTPL has not been investigated in the heavy-tailed setup.

**Adversarial Bandits with Unbounded/Heavy-tailed Losses.** In adversarial bandits, (negative) unbounded losses impose a unique challenge to ensuring the stability of a learning algorithm and hence the regret. Putta & Agrawal (2022); Huang et al. (2023a) studied "scale-free" adversarial bandits where the (deterministic) true losses could lie arbitrarily in $\mathbb{R}$. Their FTRL/Online-Mirror-Descent (OMD)-based algorithms with the log-barrier regularizer can adapt to the effective range of the losses. In the heavy-tailed setup, while the (random) losses could still be unbounded, most of the probability mass concentrates on smaller scales (around the origin). OMD with log-barrier applied to this setup can match the minimax lower bound up to a $(\ln T)^2$ factor.[2] Huang et al. (2022) proposed an FTRL-based algorithm with Tsallis entropy regularizer. While their algorithm is minimax-optimal, *an additional technical assumption is required*, which not only restricts the heavy-tailed distribution

---

[1]We use standard big $O$ notations (i.e., $O$, $\Omega$, and $\Theta$); those with a tilde (i.e., $\widetilde{O}$, $\widetilde{\Omega}$, and $\widetilde{\Theta}$) hide poly-logarithmic factors with respect to $K$ and $T$.

[2]This was suggested by Reviewer J99q due to Wei & Luo (2018, Theorem 4). We present the detailed derivations in Appendix F.

to a specific subclass but also prevents the extension to some other interesting scenarios such as our robust setting. Dorn et al. (2023) studied a slightly different setup where the heavy-tailed losses are *non-negative*, therefore the challenge from negative losses is not reflected.

**Corruption-robust Bandits/Reinforcement Learning (RL).** In this line of research, the feedback could be corrupted, while the learner's goal is still to minimize regret in the presence of corruption. There are mainly two types of corruption studied in the literature. In the first one, the data can be corrupted in total by up to some level $C$, which has been studied in both the stochastic settings (Lykouris et al., 2018; Gupta et al., 2019; He et al., 2022; Chen et al., 2021b; Lykouris et al., 2021; Wei et al., 2022) and the adversarial settings (Hajiesmaili et al., 2020; Ma & Zhou, 2023). The second model is Huber contamination (Huber, 1964), where in each round with probability $\beta$, the feedback could be corrupted arbitrarily, otherwise stays "clean". This model has been studied only in stochastic MAB (Basu et al., 2022; Wu et al., 2023) and (Contextual) Linear Bandits (Chen et al., 2022; Charisopoulos et al., 2023), whereas the adversarial settings have not been investigated. This work takes the first step towards handling Huber contamination when the losses are adversarial.

**Bandits/RL with Differential Privacy (DP).** Motivated by the need to protect users' sensitive data, lots of interest is gained in understanding the fundamental limit of bandit learning in different DP models (e.g., central model (Dwork et al., 2006), shuffle model (Cheu et al., 2019), and local model (Duchi et al., 2013)). While a large body of works study the stochastic settings (Mishra & Thakurta, 2015; Tao et al., 2022; Chowdhury & Zhou, 2022b; Shariff & Sheffet, 2018; Tenenbaum et al., 2021; Chowdhury & Zhou, 2022c; Zhou & Tan, 2021; Chowdhury & Zhou, 2022a; Qiao & Wang, 2023; Zhou, 2022), adversarial settings are much less explored (Tossou & Dimitrakakis, 2017; Agarwal & Singh, 2017). There, the proposed algorithms are FTRL-based and require additional tricks to handle unbounded noises from DP mechanisms, which leads to extra $\mathrm{polylog}(T)$ factors in the regret bound.

## 3 PROBLEM SETUP

In this section, we formally introduce the problem setup of heavy-tailed adversarial bandits. There are $K \geqslant 2$ actions and $T \geqslant K$ rounds for the learner-adversary interaction. Both $K$ and $T$ are known to the learner a priori. Before the game starts, the (oblivious) adversary determines all the loss distributions $\{P_{t,i}\}_{t\in[T],i\in[K]}$,[3] all of which satisfy the following assumption.

**Assumption 1** (Heavy-Tailed Losses)**.** The $\alpha$-th (raw) moment of all loss distributions are bounded by $\sigma^\alpha$ for some constants $\alpha \in (1,2]$ and $\sigma > 0$, i.e., $\mathbb{E}_{\ell\sim P_{t,i}}[|\ell|^\alpha] \leqslant \sigma^\alpha, \forall t \in [T], i \in [K]$.

Parameters $\alpha$ and $\sigma$ are also revealed. Then, all losses are generated according to the distributions, i.e., $\ell_{t,i} \sim P_{t,i}$, followed by the sequential interaction. In each round $t \in [T]$, the learner chooses one of the $K$ actions (denoted by $a_t$), suffers the loss of it (i.e., $\ell_{t,a_t}$), and observes $\ell_{t,a_t}$ (only).

The loss mean under $P_{t,i}$ is denoted by $\mu_{t,i} := \mathbb{E}_{\ell\sim P_{t,i}}[\ell]$. The set of all best fixed actions in hindsight is denoted by $I^* := \mathrm{argmin}_{i\in[K]} \sum_{t=1}^T \mu_{t,i}$ and $i^*$ denotes an arbitrary action in $I^*$.

The learner's goal is to minimize the *(expected pseudo-)regret*, denoted by $R_T$, which is defined as

$$R_T := \mathbb{E}\left[\sum_{t=1}^T (\mu_{t,a_t} - \mu_{t,i^*})\right], \tag{1}$$

where the randomness is from both loss generation and the learning algorithm. Regret measures the gap between the cumulative losses suffered by the learner and the best fixed action in hindsight.

*Remark* 1. Our setup is "adversarial" since the heavy-tailed distributions can change over time (in contrast to the stochastic setting in Wu et al. (2023)). Compared to the standard adversarial MAB, the adversary chooses (heavy-tailed) "distributions" rather than (deterministic) bounded losses. And this is the same setup as in the state-of-the-art (SOTA) work of Huang et al. (2022).

---

[3]The notation $[N]$ denotes the set $\{1,\ldots,N\}$ for any integer $N \geqslant 1$.

## 4 MAIN RESULTS

We first state our main theoretical result below, i.e., the regret guarantee of our proposed algorithm (Algorithm 1) under the heavy-tailed setup. A detailed description of it is presented later in Section 6.

**Theorem 1** (Regret Guarantee of Algorithm 1). *Under the setup in Section 3, Algorithm 1 achieves* $R_T = O(\sigma K^{1-\frac{1}{\alpha}} T^{\frac{1}{\alpha}} \sqrt{\ln K}), \forall \sigma > 0, \alpha \in (1, 2].$

*Remark* 2. For completeness, we provide a lower bound of $\Omega(\sigma K^{1-\frac{1}{\alpha}} T^{\frac{1}{\alpha}})$ with complete proofs in Appendix B.4.2, based on the one from Bubeck et al. (2013) for $\sigma = 1$ only. The lower bound implies that our algorithm is (nearly) *minimax-optimal*. Note that the extra $\sqrt{\ln K}$ factor is due to our choice of Laplacian perturbation. Using Fréchet-type perturbation instead and removing the $\sqrt{\ln K}$ factor (Kim & Tewari, 2019; Honda et al., 2023) is a future direction.

*Remark* 3. Theorem 1 is specific to $\alpha \in (1, 2]$. For any $\alpha > 2$, one can run our algorithm as if $\alpha = 2$ (since higher-order moments imply lower-order moments) and obtain $\tilde{O}(\sigma\sqrt{KT})$ regret, which is near-optimal since the lower bound is $\Omega(\sigma\sqrt{KT})$ for any $\alpha \geqslant 2$ (Auer et al., 2002; Bubeck et al., 2013). Therefore, our algorithm handles unbounded losses with bounded $\alpha$-th moment for any $\alpha > 1$.

**Comparision with SOTA.** While an FTRL-based algorithm with Tsallis Entropy from Huang et al. (2022) minimax-optimal, the guarantee holds *only when* the following assumption is satisfied:

**Assumption 2** (Truncated Non-negativity (Section 3.2 of Huang et al. (2022))). There exists at least one action $i^* \in I^*$ such that for any $M \geqslant 0$ and $t \in [T]$, it holds that $\mathbb{E}_{\ell \sim P_{t,i^*}}[\ell \cdot \mathbb{I}\{|\ell| > M\}] \geqslant 0$.

This assumption roughly says that *for any* threshold $M$, there should be more probability mass beyond $M$ on the positive part than the negative part beyond $-M$. Such an assumption indicates that their regret guarantee holds only for a particular class of instances. More importantly, whether this assumption holds is typically unknown to the learner.

In the rest of this paper, we first present the technical challenge from negative losses and our key observation from FTPL (Section 5). Next, our algorithm design (Section 6), and the sketched analysis (Section 7), will be presented. Lastly, we will show two direct yet important applications from our main results: (i) robust setting: adversarial bandits with heavy-tailed losses and Huber contamination (Section 8); and (ii) private setting: adversarial bandits with bounded losses with LDP (Section 9).

## 5 CHALLENGES FROM NEGATIVITY AND USEFUL INSIGHTS

In this section, we discuss why negative true losses (even bounded) incur fundamental challenges in the design and analysis of FTRL, and useful insights on why FTPL can bypass (some of) those issues.

For convenience, we first define vectors $\mu_t := (\mu_{t,1}, \dots, \mu_{t,K})$, $\ell_t := (\ell_{t,1}, \dots, \ell_{t,K})$, and $\widehat{\ell}_t := (\widehat{\ell}_{t,1}, \dots, \widehat{\ell}_{t,K})$, where $\widehat{\ell}_{t,i}$ is some constructed loss estimate for true loss $\ell_{t,i}$ (due to partial feedback in bandits). We further define vector $w_t = (w_{t,1}, \dots, w_{t,K})$ in the $(K-1)$-d probability simplex (denoted by $\Delta_{[K]}$) such that $w_{t,i} := \mathbb{P}(a_t = i | \widehat{\ell}_1, \dots, \widehat{\ell}_{t-1})$,[4] i.e., the (conditional) probability of playing action $i$ given the history. We define $K$-d vector $e_i$ such that its $i$-th element is one and zero otherwise, for any $i \in [K]$. Now, the regret can be rewritten as $R_T = \mathbb{E}\left[\sum_{t=1}^T \langle w_t - e_{i^*}, \mu_t \rangle\right]$.

Recall that FTRL first explicitly obtains $w_t$ via $w_t = \operatorname{argmin}_{w \in \Delta_{[K]}} (\phi(w)/\eta_t + \langle w, \sum_{t'=1}^{t-1} \widehat{\ell}_{t'} \rangle)$, where the "regularizer" $\phi(\cdot) : \Delta_{[K]} \to \mathbb{R}$ is some convex function and $\eta_t > 0$ is the learning rate. With $w_t$, FTRL plays action $a_t \sim w_t$ and observes $\ell_{t,a_t}$. The most common way to construct loss estimates is the "importance-weighted (IW) estimator", which yields an unbiased estimate via $\widehat{\ell}_{t,i} = \mathbb{I}\{a_t = i\} \cdot \ell_{t,i}/w_{t,i}, \forall i \in [K]$, where $\mathbb{I}\{\cdot\}$ is the indicator for a (random) event.

In general, an important property for an adversarial bandit algorithm to yield a desirable regret guarantee is the "*(single-step) stability*", meaning that $w_{t-1}$ and $w_t$ are "close". Take a common regularizer Shannon Entropy (which corresponds to the well-known EXP3/Hedge algorithm) as an example, FTRL equipped with it requires the condition that $\eta_t \cdot \widehat{\ell}_{t,i} \geqslant -1$, otherwise the

---

[4]We make the conventions that $\widehat{\ell}_{t_1}, \dots, \widehat{\ell}_{t_2}$ be $\emptyset$ (empty set) and $\sum_{t'=t_1}^{t_2} \widehat{\ell}_{t',i} = 0$ whenever $t_1 > t_2$.

standard Hedge-type regret guarantee (Dai et al., 2023, Appendix C.1) does not hold. Suppose $\widehat{\ell}_{t,i}$ is constructed via the IW estimator, then obviously this condition always holds for non-negative $\ell_{t,i}$. However, when the $\ell_{t,i}$ is a negative value with even a very small scale, the loss estimate could be still "extremely-negative" when $w_{t,i}$ is very small and could hence break the condition as well as the regret guarantee. Tsallis Entropy, the regularizer used in the SOTA work of Huang et al. (2022), faces the same issue: while arbitrary non-negative true losses ensure stability, even very small-scale negative losses could break it.[5] Therefore, whenever negative loss estimates are encountered in FTRL, much effort is needed to handle them, which leads to worse regret bounds and/or complex design/analysis.

To our best understanding, this issue is also why an artificial assumption (Assumption 2) is proposed in Huang et al. (2022). Specifically, they "truncate" overly-large losses $|\ell_{t,a_t}|$ to the scale of $O((w_{t,a_t})^{1/\alpha})$ (intuitively, to "cancel" some inversed $w_{t,a_t}$ in IW estimates) so that the stability and the regret of FTRL is nicely controlled. However, part of the total regret (contributed from such "truncation") can only be bounded by $O(\sum_{t=1}^{T}(w_{t,i^*})^{1/\alpha-1})$, which is out of control in general since $w_{t,i^*}$ could be very small. Assumption 2 ensures that this part is at most zero and hence dropped.

While such an issue exists in prior FTRL-based work, we come up with our key observation, namely a stability lemma from the literature under the FTPL framework as below.

**Lemma 1** (Single-Step Stability of FTPL, Lemma 3 of Dai et al. (2022)). *FTPL with Laplace perturbation of parameter $\eta$ ensures that $w_{t+1,i} \geqslant w_{t,i} \exp\left(-\eta \left\|\widehat{\ell}_t\right\|_1\right), \forall i \in [K], t \in [T], \eta > 0$.*

This result seems exciting and promising in the sense that, the stability prefers equally between negative losses and positive ones since it is directly controlled by $\left\|\widehat{\ell}_t\right\|_1$, i.e., the scale of loss estimates, regardless of the sign. It is somewhat intuitive since the Laplace perturbation is already a symmetric two-sided distribution and hence should not be biased toward any single side as in FTRL.

With this key lemma, our FTPL-based algorithm is shown to enjoy (nearly) optimal regret *without the need for any additional assumptions*. This advantage over FTRL brings two benefits: (i) in the robust setting, bypassing Assumption 2 allows the regret guarantee to hold against any unknown corruption distribution (which is inherent in Huber contamination model), and (ii) in the private setting, this leads to both simpler algorithm design and better regret bound upon previous works. More importantly, this lemma indicates that under the FTPL framework, there is no need to artificially control/avoid negative loss estimates, and it instead suffices to directly control the absolute value.

## 6 ALGORITHM DESIGN

Now we show how to leverage the insights above for our algorithm design. In particular, we only need an additional "skipping" trick (to avoid overly-large losses) upon the standard FTPL framework, without any other tricks or assumptions. That is, whenever the observed true loss $\ell_{t,a_t}$ has an absolute value larger than $r$ (which is non-adaptive and determined in the beginning), we simply set the loss estimate vector as a zero vector and proceed to the next round. The pseudo-code is given in Algorithm 1 and the detailed description of it is stated below.

Our algorithm takes the number of actions $K$, time horizon $T$, and heavy tail parameters $\sigma$ and $\alpha$ as the input (Line 1). Based on these, the algorithm is initialized with the Laplace distribution parameter[6] $\eta = \sigma^{-1}K^{\frac{1}{\alpha}-1}T^{-\frac{1}{\alpha}}\sqrt{\ln K}$ and skipping threshold $r = \sigma T^{\frac{1}{\alpha}}K^{-\frac{1}{\alpha}}$ (Line 2). In each round $t$, the algorithm first generates a random vector $z_t = (z_{t,1}, \ldots, z_{t,K})$ such that each element is an *i.i.d.* sample from lap($\eta$) (Line 4). Given the perturbation, the algorithm plays the "perturbed leader", which minimizes the "perturbed cumulative losses" $(z_{t,i} + \sum_{t'=1}^{t-1}\widehat{\ell}_{t',i})$ among actions $i \in [K]$ and receives bandit feedback (Lines 5 and 6). If $|\ell_{t,a_t}|$ is greater than threshold $r$, this round is "skipped" as explained in the previous paragraph (Lines 7 and 8). Otherwise, it starts the Geometric Resampling (GR) procedure to construct the loss estimates. In round $t$, counter $M_t$ counts the runs of GR (Line 15), and the maximum run is set to be $L_t = \max\{T^{1-1/\alpha}K^{1/\alpha}/e, \frac{(1-1/\alpha)\ln(K/T)}{\ln(1-\exp(-\eta\sum_{t'=1}^{t-1}\|\widehat{\ell}_{t'}\|_1)/K)}\}$ (Line 11).

---

[5] Log-barrier still enjoys $\widetilde{O}(\sqrt{T})$ dependence, but typically with an extra polylog($T$) factor.

[6] Laplace distribution with parameter $\eta$ (denoted by Lap($\eta$)) has Probability Density Function $f_\eta(x) = \eta \cdot \exp(-\eta|x|)/2, \forall x \in \mathbb{R}$.

---

**Algorithm 1** FTPL with Skipping for Heavy-Tailed Losses

---

1: **Input:** Number of actions $K$; time horizon $T$; heavy tail parameters $\sigma > 0$ and $\alpha \in (1, 2]$
2: **Initialization:** Determine Laplace distribution parameter $\eta = \sigma^{-1} K^{\frac{1}{\alpha}-1} T^{-\frac{1}{\alpha}} \sqrt{\ln K}$; determine skipping threshold $r = \sigma T^{\frac{1}{\alpha}} K^{-\frac{1}{\alpha}}$
3: **for** $t = 1 : T$ **do**
4:      Sample perturbation $z_t$ such that $z_{t,i} \overset{\text{i.i.d.}}{\sim} \text{Lap}(\eta), \forall i \in [K]$
5:      Play action $a_t = \text{argmin}_{i \in [K]} \left( z_{t,i} + \sum_{t'=1}^{t-1} \widehat{\ell}_{t',i} \right)$
6:      Observe $\ell_{t,a_t}$
7:      **if** $|\ell_{t,a_t}| > r$ **then**
8:          Construct loss estimate $\widehat{\ell}_t = \mathbf{0}$
9:      **else**
10:          GR counter $M_t = 0$
11:          Set GR maximum runs as $L_t = \max\{T^{1-\frac{1}{\alpha}} K^{\frac{1}{\alpha}}/e, \frac{(1-1/\alpha)\ln(K/T)}{\ln\left(1-\exp(-\eta \sum_{t'=1}^{t-1} \|\widehat{\ell}_{t'}\|_1)/K\right)}\}$
12:          **while** $M_t \leqslant L_t - 1$ **do**
13:              Sample a fresh perturbation $\tilde{z}$ in the same way as $z_t$
14:              Calculate $a'_t = \text{argmin}_{i \in [K]} \left( \tilde{z}_i + \sum_{t'=1}^{t-1} \widehat{\ell}_{t',i} \right)$
15:              Update GR counter $M_t = M_t + 1$
16:              **if** $a'_t = a_t$ **then**
17:                  BREAK, i.e., go to Line 20
18:              **end if**
19:          **end while**
20:          Construct loss estimate $\widehat{\ell}_{t,i} = \mathbb{I}\{a_t = i\} M_t \ell_{t,i}, \forall i \in [K]$
21:      **end if**
22: **end for**

---

Finally, a While-Loop is executed to obtain $M_t$: in each iteration, a new perturbation $\tilde{z}$ is sampled from fresh randomness in the same way as $z_t$ (Line 13) and the "perturbed leader" $a'_t$ with respect to (w.r.t.) $\tilde{z}$ is calculated (Line 14). If $a'_t$ is the action played in the current round (i.e., $a_t$), the While-Loop is broken (Lines 16 and 17). Otherwise, it proceeds to the next iteration, unless the maximum run is reached. The loss estimate is constructed by $M_t$ (the number of GR runs) multiplying the true loss for action $a_t$, and zero for the others (Line 20).

*Remark* 4. In general, it is difficult to obtain the closed-form expression of $w_t$ in FTPL. Hence, the IW estimator is not applicable (Abernethy et al., 2016), and GR is proposed as an alternative. The purpose of setting the maximum GR runs is to ensure a *finite* and *deterministic* running time upper bound (Neu & Bartók, 2016) (otherwise it is possible that in a single run, the While-Loop may never stop), which incurs additional regret due to biasedness. However, this part is negligible for sufficiently large $L_t$, whose choice is discussed in Section 7.3.

## 7    ANALYSIS SKETCH

In this section, we provide the sketched analysis for the regret guarantee of our Algorithm 1. All omitted proofs in this section are given in Appendix B. We will first show a general decomposition of the regret into three terms and then bound each term separately, all of order $\widetilde{O}(\sigma K^{1-1/\alpha} T^{1/\alpha})$. The decomposition is formally stated in the lemma below, obtained by simply rewriting the regret.

**Lemma 2** (Regret Decomposition)**.** *The regret can be decomposed as*

$$R_T = \underbrace{\mathbb{E}\left[\sum_{t=1}^{T} \langle w_t - e_{i^*}, \mu_t - \mu'_t \rangle\right]}_{\text{SKIPERR}} + \underbrace{\mathbb{E}\left[\sum_{t=1}^{T} \langle w_t - e_{i^*}, \widehat{\ell}_t \rangle\right]}_{\text{FTPLREG}} + \underbrace{\mathbb{E}\left[\sum_{t=1}^{T} \langle w_t - e_{i^*}, \mu'_t - \widehat{\ell}_t \rangle\right]}_{\text{GRERR}},$$

*where* $\mu'_t := (\mu'_{t,1}, \ldots, \mu'_{t,K})$, *and* $\mu'_{t,i} := \mathbb{E}_{\ell \sim P_{t,i}}[\ell \cdot \mathbb{I}\{|\ell| \leqslant r\}], \forall t \in [T], i \in [K]$.

*Remark* 5. SKIPERR (standing for "Skipping Error") can be viewed as the error from skipping, which goes to zero without skipping ($r \to \infty$). FTPLREG (standing for "FTPL Regret") is the regret w.r.t.

loss sequence $\widehat{\ell}_1, \ldots, \widehat{\ell}_T$. GRERR (standing for "GR Error") comes from setting the maximum runs of GR and yielding biased loss estimates, which goes to zero with unbiased estimates ($L_t \to \infty$).

## 7.1 BOUNDING THE SKIPPING ERROR

To bound Skipping Error, we follow the analysis from Huang et al. (2022, Lemmas A.2 and A.3). Specifically, for any fixed $r > 0$, the Skipping Error could be bounded by $2T\sigma^\alpha r^{1-\alpha}$. Therefore, our modified choice of $r = \sigma T^{1/\alpha} K^{-1/\alpha}$ leads to a bound of $2\sigma K^{1-1/\alpha} T^{1/\alpha}$.

## 7.2 BOUNDING THE FTPL REGRET

In summary, we will follow the standard analysis to decompose the FTPL Regret into two terms, respectively named "Error Term" (denoted by ERRTERM) and "Stability Term" (denoted by STATERM), and then bound them separately.

**Lemma 3** (FTPL Regret Decomposition. Lemma 3 of Honda et al. (2023))**.** *Algorithm 1 ensures that*

$$\text{FTPLREG} = \mathbb{E}\left[\sum_{t=1}^{T}\langle w_t - e_{i^*}, \widehat{\ell}_t\rangle\right] \leqslant 2 \cdot \underbrace{\mathbb{E}_{\bar{z}_i \overset{i.i.d.}{\sim} Lap(\eta)}\left[\max_{i\in[K]}|\bar{z}_i|\right]}_{\text{ERRTERM}} + \underbrace{\mathbb{E}\left[\sum_{t=1}^{T}\langle \widehat{\ell}_t, w_t - w_{t+1}\rangle\right]}_{\text{STATERM}}, \forall \eta > 0, r > 0.$$

*Remark* 6. This lemma reflects that the Stability Term (and hence FTPL Regret) could be nicely controlled if $w_t$ and $w_{t+1}$ are close, which is exactly the desired "stability" property we refer to.

### 7.2.1 BOUNDING THE ERROR TERM

The Error Term is just the (doubled) expected largest absolute value among $K$ i.i.d. samples from $Lap(\eta)$, which is bounded by $6\ln(K)/\eta$ given Fact 2 of Wang & Dong (2020) (also see Lemma 10).

### 7.2.2 BOUNDING THE STABILITY TERM

The key for this part is the "single-step stability" lemma mentioned in Section 5, and we restate it below for convenience.

**Lemma 4.** *Algorithm 1 ensures that* $w_{t+1,i} \geqslant w_{t,i}\exp\left(-\eta\left\|\widehat{\ell}_t\right\|_1\right), \forall i \in [K], t \in [T], \eta > 0.$

*Remark* 7. The proof of Lemma 4 relies heavily on the fact that the perturbations are sampled from fresh randomness (so that they are still Laplacian conditioned on the history). While Syrgkanis et al. (2016) showed that with *full information* (where the losses of all actions are revealed) against *oblivious adversaries*, it suffices to use the fixed perturbation sampled in the first round, it is unclear whether this is achievable in bandit feedback. Therefore, the proposed fixed perturbations for the bandit case in Dai et al. (2022) should be replaced with fresh perturbations. Accordingly, bounding the Error Term (their Lemma 10), which relies heavily on fixed perturbations, may need a revisit.

Given the single-step stability, it is left to bound the entire Stability Term, adapting the proof of Dai et al. (2022, Lemma 12) to our heavy-tailed setup. The upper bound is stated in the lemma below.

**Lemma 5.** *Algorithm 1 ensures that* STATERM $\leqslant 2\eta\sigma^\alpha r^{2-\alpha} KT, \forall \eta > 0, r > 0.$

*Remark* 8. The role of skipping here is different from Huang et al. (2022). Specifically, our skipping simply helps avoid overly-large losses (w.r.t. a fixed threshold $r$), and theirs is to "cancel" some $w_{t,a_t}$, while their skipping threshold (of order $\Theta((w_{t,a_t})^{1/\alpha})$) *is not necessarily large* (for small $w_{t,a_t}$).

## 7.3 BOUNDING THE GR ERROR

This subsection focuses on the GR Error. The lemma below formally states an upper bound on it.

**Lemma 6.** *With $r$ and $L_t$ specified in Algorithm 1, it is ensured that* GRERR $\leqslant 2\sigma K^{1-1/\alpha} T^{1/\alpha}.$

*Proof Sketch of Lemma 6.* To show this, we first decompose the GR Error into two parts by

$$\text{GRERR} = \underbrace{\mathbb{E}\left[\sum_{t=1}^{T}\langle w_t, \mu'_t - \widehat{\ell}_t\rangle\right]}_{\text{GRERR I}} + \underbrace{\mathbb{E}\left[\sum_{t=1}^{T}\langle e_{i^*}, \widehat{\ell}_t - \mu'_t\rangle\right]}_{\text{GRERR II}}.$$

While it is standard to choose $L_t \geqslant K^{1/\alpha}T^{1-1/\alpha}/e, \forall t \in [T]$ so that GRERR I $\leqslant \sigma K^{1-1/\alpha}T^{1/\alpha}$ (Neu & Bartók, 2016), the challenge arises in the second term. Specifically, while GRERR II could be simply bounded by zero when the losses are *non-negative*, it is not easy to control as the first term when negative losses are allowed and we are not aware of any existing solution. As a workaround, we first notice that GRERR II can be bounded by $\mathbb{E}[\sigma \sum_{t=1}^{T}(1 - w_{t,i^*})^{L_t}]$ which implies the key is to obtain a lower bound of $w_{t,i^*}$ (so that choosing sufficiently large $L_t$ suffices). Now, by applying Lemma 4 recursively, we show that $w_{t,i} \geqslant \exp\left(-\eta \sum_{t'=1}^{t-1} \left\|\widehat{\ell}_{t'}\right\|_1\right)/K, \forall t \in [T], i \in [K]$, which allows us to set sufficiently large (but still finite) $L_t$ and to bound GRERR II by $\sigma K^{1-1/\alpha}T^{1/\alpha}$. $\quad\square$

Finally, putting the respective upper bounds on the three terms in Lemma 2 together, along with $\eta$, $r$, and $L_t$ specified in Algorithm 1, yields the regret guarantee in Theorem 1.

## 8 HEAVY-TAILED ADVERSARIAL BANDITS WITH HUBER CONTAMINATION

In this section, we give the first application of our main result, which is adversarial bandits with heavy-tailed losses and Huber contamination, defined formally as below. All omitted details/proofs in this section are given in Appendix C.

**Definition 1** (Heavy-tailed Adversarial Bandits with Huber Contamination). In addition to the protocol given in Section 3, in each round $t$ with some probability $\beta \in [0, 1]$ (known to the learner), the loss observed by the learner (denoted by $\widetilde{\ell}_{t,a_t}$) is no longer the true loss $\ell_{t,a_t}$ generated from "clean" distribution $P_{t,a_t}$ (satisfying Assumption 1), but instead is "contaminated" via being generated from some *arbitrary and unknown* "bad" distribution $Q_{t,a_t}$. The regret is still measured w.r.t. clean losses given by $R_T := \mathbb{E}\left[\sum_{t=1}^{T}(\mu_{t,a_t} - \mu_{t,i^*})\right]$, while the expectation additionally includes the randomness from the potential contaminations.

In this setup, our Algorithm 1 with modified parameters (where $\beta$ is involved) is shown to enjoy the following upper bound. The choices of modified parameters are given in Appendix C.1.

**Lemma 7** (Regret Guarantee with Huber Contamination (Informal)). *Under the setup given by Definition 1, Algorithm 1 with modified $\eta$, $r$, and $L_t$, ensures that $R_T = \widetilde{O}(\sigma(T^{\frac{1}{\alpha}}K^{1-\frac{1}{\alpha}} + T\beta^{1-\frac{1}{\alpha}}))$.*

*Proof Sketch of Lemma 7.* We first note that the observed loss $\tilde{\ell}_{t,i}$ could be viewed as a sample directly from "mixed distribution" $P_{\beta,t,i} := (1 - \beta)P_{t,i} + \beta Q_{t,i}$. Moreover, we will ignore the GR Error term (i.e., let $L_t \to \infty$) since it can be bounded following the same steps as in Section 7.3 with a modified $L_t$. Now we can decompose the regret by

$$R_T = \underbrace{\mathbb{E}\left[\sum_{t=1}^{T}\langle w_t - e_{i^*}, \mu_t - \mu_t''\rangle\right]}_{\text{SKIPERR}'} + \underbrace{\mathbb{E}\left[\sum_{t=1}^{T}\langle w_t - e_{i^*}, \widehat{\ell}_t\rangle\right]}_{\text{FTPLREG}'},$$

where $\mu_t'' := (\mu_{t,1}'', \ldots, \mu_{t,K}'')$ and $\mu_{t,i}'' := \mathbb{E}_{\ell \sim P_{\beta,t,i}}[\ell \cdot \mathbb{I}\{|\ell| \leqslant r\}], \forall i \in [K], t \in [T]$. We obtain the final regret by bounding these two terms separately and then choosing $\eta$ and $r$ accordingly. $\quad\square$

*Remark* 9. We provide a matching lower bound in Appendix C.2, which implies that our algorithm is (nearly) *minimax-optimal*. This is *the first non-trivial (and indeed optimal) regret guarantee in adversarial environments*, which together with Wu et al. (2023) for the stochastic case indicates that in this setup, adversarial environments is *no harder than* stochastic ones (in terms of minimax regret). While the second term in the regret is linear in $T$, we note that this is a common pattern due to arbitrary contaminations (Chen et al., 2022), and the key is the optimal $\Theta(\beta^{1-\frac{1}{\alpha}})$ dependence.

*Remark* 10. With *unknown and arbitrary* bad distributions (as in Huber model), the FTRL-based algorithm in Huang et al. (2022) with the current analysis cannot provide any non-trivial regret guarantee in general, because the needed Assumption 2 (which now should be modified as $\mathbb{E}_{\ell \sim P_{t,i^*}}[\ell] - \mathbb{E}_{\ell \sim P_{\beta,t,i^*}}[\ell \cdot \mathbb{I}\{|\ell| \leqslant r\}] \geqslant 0$) in the worst case does not hold, while our regret guarantee holds for any clean distributions and bad ones.

# 9 ADVERSARIAL BANDITS WITH BOUNDED LOSSES AND LOCAL DIFFERENTIAL PRIVACY

Another direct yet important application of our result is the standard adversarial bandits (i.e., losses are deterministic and bounded in $[0, 1]$) with LDP, which has also been studied in Agarwal & Singh (2017); Tossou & Dimitrakakis (2017).[7] All details/proofs are given in Appendix D. We first give the definition of DP followed by the learning setup.

**Definition 2** (Differential Privacy (DP)). For any given privacy budget $\varepsilon > 0$, a mechanism $\mathcal{M} : \mathcal{D} \to \mathbb{R}^m$ is said to be $\varepsilon$-differentially private (DP) if for all datasets $X, X'$ in $\mathcal{D}$ that differ on only one element and measurable subset $\mathcal{E} \subset \mathbb{R}^m$, it holds that $\mathbb{P}(M(X) \in \mathcal{E}) \leqslant \exp(\varepsilon) \cdot \mathbb{P}(M(X') \in \mathcal{E})$.

**Definition 3** (Adversarial Bandits with Bounded Losses and LDP). True losses $\ell_1, \ldots, \ell_T \in [0, 1]^K$ are deterministic and chosen by an oblivious adversary. Given any privacy budget $\varepsilon \in (0, 1]$, the bandit model is said to be $\varepsilon$-LDP if $a_{t+1}$ lies in the sigma-algebra generated by $\{a_{t'}, M(\ell_{t,a_{t'}})\}_{t' \in [t]}$ in any round $t \in [T]$ where $M$ is an $\varepsilon$-DP mechanism.

Roughly speaking, the algorithm should not touch true losses, and it observes privatized losses only. Here, we adopt the widely-used Laplace mechanism (Dwork et al., 2014). Specifically, when data are bounded in $[0, 1]$, adding noise drawn from $\text{Lap}(\varepsilon)$ to them ensures $\varepsilon$-DP. By adopting it, the observed loss is the true loss plus an i.i.d. sample from $\text{Lap}(\varepsilon)$, which is shown to satisfy Assumption 1 with $\sigma = \Theta(1/\varepsilon)$ and $\alpha = 2$. That is, this setup could be viewed as a specific way of generating heavy-tailed losses (i.e., bounded true loss + Laplace noise for privacy). Plugging $\sigma = \Theta(1/\varepsilon)$ and $\alpha = 2$ into Thorem 1, we directly obtain the regret guarantee in the LDP model, which is formally stated below.

**Corollary 1** (Regret Guarantee under $\varepsilon$-LDP). *Given privacy budget $\varepsilon \in (0, 1]$, Algorithm 1 with Laplace mechanism, $\sigma = \Theta(1/\varepsilon)$, and $\alpha = 2$ guarantees both $\varepsilon$-LDP and $R_T = O(\sqrt{KT \ln K}/\varepsilon)$.*

*Remark* 11. Our result improves Agarwal & Singh (2017, Theorem 4.1) by a $\sqrt{\ln T}$ factor and Tossou & Dimitrakakis (2017, Corollary 3.1) by a $\ln T$ factor, both of which are FTRL-based with explicit uniform exploration (which is unnecessary in FTPL) to ensure stability. Given the lower bound in Garcelon et al. (2021, Theorem 2), our upper bound is tight (up to a $\sqrt{\ln K}$ factor) for all $\varepsilon \leqslant 1$.

*Remark* 12. As emphasized before, a subtle but central issue here is the negative (privatized) loss. That is, solely injecting unbounded private noise in FTRL-based algorithms (for losses in $[0, 1]$), may break the stability as well as the regret guarantee. This is the motivation behind the additional use of explicit exploration in FTRL from prior works (i.e., to ensure stability). As a (counter-)example, Zheng et al. (2020, Theorem 11) claimed a best-of-both-words (BOBW)[8] regret bound with LDP guarantee by simply injecting Gaussian noise in an FTRL/OMD-based optimal BOBW algorithm for bounded losses from Zimmert & Seldin (2021). It is unclear whether their claimed bounds indeed hold, since the original analysis *relies heavily on non-negative* observed losses, which no longer holds with Gaussian noise.

# 10 CONCLUSION

We propose an FTPL-based learning algorithm for heavy-tailed adversarial MAB, and show that it enjoys a near-optimal worst-case regret upper bound without the need for additional undesired assumptions in prior works. We also present two important applications of our algorithm—one robust setting and one private setting—and show that our algorithm enjoys either the first optimal regret guarantee or improved results with an even simpler design (compared to FTRL-based ones). The key insight behind the improvements lies in an existing stability lemma from FTPL. Future directions include (i) exploring more problem setups where FTPL could be advantageous over FTRL and (ii) jointly considering DP and Huber robustness (Chhor & Sentenac, 2023; Li et al., 2023; Hopkins et al., 2023; Wu et al., 2023; Asi et al., 2023) in various Online Learning problems.

---

[7]While they claimed (central) DP, their algorithms actually ensures (the stronger) LDP which implies DP.

[8]BOBW refers to that one single algorithm can achieve $\sqrt{T}$-type regret in adversarial environments and $\text{polylog}\, T$-type regret in stochastic environments, *without being aware of* the underlying environment type.

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

# Appendix

## A  ADDITIONAL RELATED WORK

**Follow-the-Regularized-Leader (FTRL).** FTRL is probably the most celebrated class of algorithms for solving Online Learning problems. The key idea is to be greedy with respect to (w.r.t.) the cumulative (estimated) losses so far but with an additive regularization term to ensure stability. As a generic algorithmic framework, it captures many well-known (bandit) algorithms as special cases, such as EXP3 (with Negative Shannon Entropy being the regularizer) (Littlestone & Warmuth, 1989) and Implicitly Normalized Forecaster or INF (with $1/2$-Tsallis Entropy being the regularizer) (Audibert & Bubeck, 2009). Due to its relatively simple recipe for theoretical analyses, FTRL has been applied to a broad range of Online Learning problems with bandit feedback, including Linear Bandits (Bartlett et al., 2008), bandits with graph feedback (Mannor & Shamir, 2011; Luo et al., 2023), (episodic) tabular Markov Decision Process (MDPs) (Jin et al., 2020), Linear MDPs (Dai et al., 2023), and Best-of-Both-World (BoBW) guarantees in various settings (Zimmert & Seldin, 2021; Jin et al., 2021; 2023).

**Heavy Tails in Stochastic Bandits/Reinforcement Learning (RL).**  "Stochastic" refers to the setting where the loss distributions remain fixed throughout the game. Heavy-tailed stochastic losses has a large body of literature, originating from MAB in Bubeck et al. (2013) and expanding to Linear Bandits (Medina & Yang, 2016; Shao et al., 2018; Zhong et al., 2021; Xue et al., 2021; Kang & Kim, 2023), Kernelized/Gaussian-process Bandits (Ray Chowdhury & Gopalan, 2019), episodic tabular MDPs (Zhuang & Sui, 2021), and Linear MDPs (Huang et al., 2023b; Li & Sun, 2023). The common idea for handling heavy-tailed stochastic losses is to derive mean estimation concentration results via robust estimators including median-of-means, truncated mean, Catoni's M-estimator (Catoni, 2012; Chen et al., 2021a; Bhatt et al., 2022), and Huber's estimator (Huber, 1964; Sun et al., 2020).

## B  OMITTED DETAILS IN SECTION 7

### B.1  OMITTED DETAILS IN SECTION 7.1 (BOUNDING SKIPPING ERROR)

**Lemma 8.** *For any skipping threshold $r > 0$, we have $|\mu_{t,i} - \mu'_{t,i}| \leqslant \sigma^\alpha r^{1-\alpha}, \forall i \in [K], t \in [T]$.*

*Proof of Lemma 8.*  Based on the definitions of $\mu_{t,i}$ and $\mu'_{t,i}$, we have

$$
\begin{aligned}
\mu_{t,i} - \mu'_{t,i} &= \mathop{\mathbb{E}}_{\ell \sim P_{t,i}}[\ell] - \mathop{\mathbb{E}}_{\ell \sim P_{t,i}}[\ell \cdot \mathbb{I}\{|\ell| \leqslant r\}] \\
&= \mathop{\mathbb{E}}_{\ell \sim P_{t,i}}[\ell \cdot \mathbb{I}\{|\ell| > r\}] \\
&\leqslant \mathop{\mathbb{E}}_{\ell \sim P_{t,i}}[|\ell| \cdot \mathbb{I}\{|\ell| > r\}] \\
&\leqslant \mathop{\mathbb{E}}_{\ell \sim P_{t,i}}[|\ell|^\alpha r^{1-\alpha} \cdot \mathbb{I}\{|\ell| > r\}] \\
&\leqslant \mathop{\mathbb{E}}_{\ell \sim P_{t,i}}[|\ell|^\alpha r^{1-\alpha}] \\
&\leqslant \sigma^\alpha r^{1-\alpha}.
\end{aligned}
$$

Similarly,

$$
\begin{aligned}
\mu'_{t,i} - \mu_{t,i} &= \mathop{\mathbb{E}}_{\ell \sim P_{t,i}}[\ell \cdot \mathbb{I}\{|\ell| \leqslant r\}] - \mathop{\mathbb{E}}_{\ell \sim P_{t,i}}[\ell] \\
&= \mathop{\mathbb{E}}_{\ell \sim P_{t,i}}[-\ell \cdot \mathbb{I}\{|\ell| > r\}] \\
&\leqslant \mathop{\mathbb{E}}_{\ell \sim P_{t,i}}[|\ell| \cdot \mathbb{I}\{|\ell| > r\}] \\
&\leqslant \sigma^\alpha r^{1-\alpha}.
\end{aligned}
$$

Combining them completes the proof. □

**Lemma 9** (Bounding Skipping Error). *For any skipping threshold $r > 0$, we have* SKIPERR $\leqslant 2T\sigma^{\alpha}r^{1-\alpha}$.

*Proof of Lemma 9.* In any single round, we have

$$\mathbb{E}\left[\langle w_t - e_{i^*}, \mu_{t,i} - \mu'_{t,i}\rangle\right] = \sum_{i=1}^{K} w_{t,i}(\mu_{t,i} - \mu'_{t,i}) + (\mu'_{t,i^*} - \mu_{t,i^*}) \leqslant \sum_{i=1}^{K} w_{t,i}\sigma^{\alpha}r^{1-\alpha} + \sigma^{\alpha}r^{1-\alpha} = 2\sigma^{\alpha}r^{1-\alpha},$$

where the inequality is from Lemma 8 and the last step is simply due to $\sum_{i=1}^{K} w_{t,i} = 1$. Taking the summation over all $T$ rounds completes the proof. □

### B.2 OMITTED DETAILS IN SECTION 7.2 (BOUNDING FTPL REGRET)

**Lemma 10** (Fact 2 of Wang & Dong (2020)). *Suppose $X_1, \ldots, X_K$ are i.i.d. samples form $Lap(\eta)$, it holds that*

$$\mathbb{E}\left[\max_{i \in [K]} |X_i|\right] \leqslant \frac{1 + \ln K}{\eta} \overset{(K \geqslant 2)}{\leqslant} \frac{3 \ln K}{\eta}.$$

*Proof of Lemma 3.* Let us look at one particular trajectory $\{w_1, \widehat{\ell}_1, \ldots, w_T, \widehat{\ell}_T\}$. Let $\bar{z} := (\bar{z}_1, \ldots, \bar{z}_K) \in \mathbb{R}^K$ be a random vector generated in the same way as all the perturbations (i.e., Line 4) and be independent to the trajectory. Now define (random) vector $u_t = (u_{t,1}, \ldots, u_{t,K})$, such that its each entry is defined by $u_{t,i} := \mathbb{I}\{i = \arg\min_{j \in [K]}(\bar{z}_j + \sum_{t'=1}^{t-1} \widehat{\ell}_{t',j})\}, \forall i \in [K]$ and hence $u_t \in \{e_i\}_{i \in [K]}$. Since $z_t$ (the perturbation generated in Line 4) and $\bar{z}$ are independent and identically distributed conditioned on the history up to round $t-1$, we have

$$\mathbb{E}[u_t|\widehat{\ell}_1, \ldots, \widehat{\ell}_{t-1}] = w_t.$$

Now we have

$$\begin{aligned}
\sum_{t=1}^{T} \langle \widehat{\ell}_t, e_{i^*}\rangle &= \left\langle \sum_{t=1}^{T} \widehat{\ell}_t + \bar{z}, e_{i^*}\right\rangle - \langle \bar{z}, e_{i^*}\rangle \\
&\overset{(a)}{\geqslant} \left\langle \sum_{t=1}^{T} \widehat{\ell}_t + \bar{z}, u_{T+1}\right\rangle - \langle \bar{z}, e_{i^*}\rangle \\
&= \left\langle \sum_{t=1}^{T-1} \widehat{\ell}_t + \bar{z}, u_{T+1}\right\rangle + \left\langle \widehat{\ell}_T, u_{T+1}\right\rangle - \langle \bar{z}, e_{i^*}\rangle \\
&\overset{(b)}{\geqslant} \left\langle \sum_{t=1}^{T-1} \widehat{\ell}_t + \bar{z}, u_T\right\rangle + \left\langle \widehat{\ell}_T, u_{T+1}\right\rangle - \langle \bar{z}, e_{i^*}\rangle,
\end{aligned}$$

where step (a) is because $u_{T+1}$ is defined to be the "leader" w.r.t. loss vector $\left(\sum_{t=1}^{T} \widehat{\ell}_t + \bar{z}\right)$, and step (b) is obtained for the same reason.

By doing this recursively, we get

$$\sum_{t=1}^{T} \left\langle \widehat{\ell}_t, e_{i^*}\right\rangle \geqslant \langle \bar{z}, u_1\rangle - \langle \bar{z}, e_{i^*}\rangle + \sum_{t=1}^{T} \langle \widehat{\ell}_t, u_{t+1}\rangle,$$

and after rearranging it, we have

$$\sum_{t=1}^{T} \left\langle \widehat{\ell}_t, u_t - e_{i^*}\right\rangle \leqslant \langle \bar{z}, e_{i^*} - u_1\rangle + \sum_{t=1}^{T} \langle \widehat{\ell}_t, u_t - u_{t+1}\rangle.$$

Now by taking expectation on both sides over the randomness from $\bar{z}$ (while the trajectory is fixed), we arrive at

$$\sum_{t=1}^{T} \langle \widehat{\ell}_t, w_t - e_{i^*} \rangle \leqslant \mathbb{E}\left[\langle \bar{z}, e_{i^*} - u_1 \rangle\right] + \sum_{t=1}^{T} \langle \widehat{\ell}_t, w_t - w_{t+1} \rangle.$$

Further taking expectations on both sides over the randomness of the trajectory, we get

$$\mathbb{E}\left[\sum_{t=1}^{T} \langle \widehat{\ell}_t, w_t - e_{i^*} \rangle\right] \leqslant \mathbb{E}\left[\langle \bar{z}, e_{i^*} - u_1 \rangle\right] + \left[\sum_{t=1}^{T} \langle \widehat{\ell}_t, w_t - w_{t+1} \rangle\right].$$

Finally, noticing that both $e_{i^*}$ and $u_1$ lie in the probability simplex, we have

$$\mathbb{E}\left[\langle \bar{z}, e_{i^*} - u_1 \rangle\right] \leqslant 2 \cdot \mathop{\mathbb{E}}_{\bar{z}_i \overset{\text{i.i.d}}{\sim} \text{Lap}(\eta)}\left[\max_{i \in [K]} |\bar{z}_i|\right],$$

which completes the proof. $\qquad\square$

*Proof of Lemma 4.* In this proof, we abuse the notation a little bit and use $f_\eta(z)$ to denote the Probability Density Function (PDF) at vector $z = (z_1, \ldots, z_K)$ such that $z_i \sim \text{Lap}(\eta), \forall i \in [K]$. We first rewrite $w_{t,i}$ as

$$w_{t,i} = \mathbb{P}(a_t = i | \widehat{\ell}_1, \ldots, \widehat{\ell}_{t-1})$$

$$\overset{(a)}{=} \int_z \mathbb{I}\{e_i = \mathop{\text{argmin}}_{w \in \{e_i\}_{i \in [K]}} \langle \sum_{t'=1}^{t-1} \widehat{\ell}_{t'} + z, w \rangle\} f_\eta(z) \, dz$$

$$\overset{(b)}{=} \int_z \mathbb{I}\{e_i = \mathop{\text{argmin}}_{w \in \{e_i\}_{i \in [K]}} \langle \sum_{t'=1}^{t-1} \widehat{\ell}_{t'} + (z + \widehat{\ell}_t), w \rangle\} f_\eta(z + \widehat{\ell}_t) \, d(z + \widehat{\ell}_t)$$

$$\overset{(c)}{=} \int_z \mathbb{I}\{e_i = \mathop{\text{argmin}}_{w \in \{e_i\}_{i \in [K]}} \langle \sum_{t'=1}^{t} \widehat{\ell}_{t'} + z, w \rangle\} f_\eta(z + \widehat{\ell}_t) \, dz,$$

where step (a) is due to the fact that perturbation is sampled from fresh randomness (and hence $z$ is still Laplacian conditioned on the history), in step (b) we use the "change of variables" trick and the fact that $z$ has support over the whole $\mathcal{R}^K$ space, and step (c) is because $(z + \widehat{\ell}_t)$ is linear in $z$.

Similarly, we can write $w_{t+1,i}$ as

$$w_{t+1,i} = \int_z \mathbb{I}\{e_i = \mathop{\text{argmin}}_{w \in \{e_i\}_{i \in [K]}} \langle \sum_{t'=1}^{t} \widehat{\ell}_{t'} + z, w \rangle\} f_\eta(z) \, dz.$$

Since each element in $z$ is an i.i.d sample from $\text{Lap}(\eta)$, we have

$$f_\eta(z) = \prod_{i \in [K]} \frac{\eta}{2} \exp(-\eta|z_i|) = \left(\frac{\eta}{2}\right)^K \exp(-\eta \|z\|_1),$$

and similarly we get

$$f_\eta(z + \widehat{\ell}_t) = \left(\frac{\eta}{2}\right)^K \exp\left(-\eta\|z + \widehat{\ell}_t\|_1\right).$$

Taking the ratio between them, by triangle inequality, we obtain

$$\frac{f_\eta(z + \widehat{\ell}_t)}{f_\eta(z)} = \exp\left(-\eta\left(\|z + \widehat{\ell}_t\|_1 - \|z\|_1\right)\right) \in \left[\exp\left(-\eta\|\widehat{\ell}_t\|_1\right), \exp\left(\eta\|\widehat{\ell}_t\|_1\right)\right].$$

Since this ratio of the two densities holds for any $z$ in $\mathbb{R}^K$, it also applies to the integrals (i.e., $w_{t+1,i}/w_{t,i}$), which completes the proof. $\qquad\square$

*Proof of Lemma 5.* Based on Lemma 4, for any round $t \in [T]$ we have

$$
\begin{aligned}
\sum_{i=1}^{K} (w_{t,i} - w_{t+1,i}) \widehat{\ell}_{t,i} &\leqslant \sum_{i=1}^{K} w_{t,i} \left(1 - \exp(-\eta \left\| \widehat{\ell}_t \right\|_1)\right) \widehat{\ell}_{t,i} \\
&\overset{(a)}{\leqslant} \sum_{i=1}^{K} w_{t,i} \left(\eta \left\| \widehat{\ell}_t \right\|_1\right) \widehat{\ell}_{t,i} \\
&\overset{(b)}{=} \eta \sum_{i=1}^{K} w_{t,i} (\widehat{\ell}_{t,i})^2,
\end{aligned}
$$

where step (a) is from elementary inequality $1 - \exp(-x) \leqslant x$, and step (b) is due to the fact that $\left\| \widehat{\ell}_t \right\|_1 \cdot \widehat{\ell}_{t,i} = (\widehat{\ell}_{t,i})^2$ since $\widehat{\ell}_t$ has at most one non-zero entry.

Taking conditional expectations on both sides, we have

$$
\begin{aligned}
\mathbb{E}\left[\sum_{i=1}^{K}(w_{t,i} - w_{t+1,i})\widehat{\ell}_{t,i}|\widehat{\ell}_1, \ldots, \widehat{\ell}_{t-1}\right] &\leqslant \eta \cdot \mathbb{E}\left[\sum_{i=1}^{K} w_{t,i}(\widehat{\ell}_{t,i})^2|\widehat{\ell}_1, \ldots, \widehat{\ell}_{t-1}\right] \\
&= \eta \sum_{i=1}^{K} w_{t,i}\mathbb{E}[(\mathbb{I}\{a_t = i\}\mathbb{I}\{|\ell_{t,i}| \leqslant r\}M_t\ell_{t,i})^2|\widehat{\ell}_1, \ldots, \widehat{\ell}_{t-1}] \\
&= \eta \sum_{i=1}^{K} w_{t,i}\mathbb{E}[\mathbb{I}\{a_t = i\}\mathbb{I}\{|\ell_{t,i}| \leqslant r\}(M_t)^2(\ell_{t,i})^2|\widehat{\ell}_1, \ldots, \widehat{\ell}_{t-1}] \\
&\leqslant \eta \sum_{i=1}^{K} w_{t,i}\mathbb{E}[\mathbb{I}\{a_t = i\}(M_t)^2(\ell_{t,i})^{\alpha}r^{2-\alpha}|\widehat{\ell}_1, \ldots, \widehat{\ell}_{t-1}] \\
&\leqslant \eta\sigma^{\alpha}r^{2-\alpha} \sum_{i=1}^{K} w_{t,i}\mathbb{E}[\mathbb{I}\{a_t = i\}(M_t)^2|\widehat{\ell}_1, \ldots, \widehat{\ell}_{t-1}] \\
&= \eta\sigma^{\alpha}r^{2-\alpha} \sum_{i=1}^{K} w_{t,i} \cdot \mathbb{P}\left(a_t = i|\widehat{\ell}_1, \ldots, \widehat{\ell}_{t-1}\right) \cdot \mathbb{E}\left[(M_t)^2|\widehat{\ell}_1, \ldots, \widehat{\ell}_{t-1}, a_t = i\right] \\
&= \eta\sigma^{\alpha}r^{2-\alpha} \sum_{i=1}^{K} (w_{t,i})^2 \mathbb{E}\left[(M_t)^2|\widehat{\ell}_1, \ldots, \widehat{\ell}_{t-1}, a_t = i\right] \\
&\overset{(a)}{\leqslant} \eta\sigma^{\alpha}r^{2-\alpha} \sum_{i=1}^{K} (w_{t,i})^2 \left(\frac{1 - w_{t,i}}{(w_{t,i})^2} + \frac{1}{(w_{t,i})^2}\right) \\
&= 2\eta\sigma^{\alpha}r^{2-\alpha}K,
\end{aligned}
$$

where step (a) follows from Lemma 12. Taking the summation over $T$ rounds completes the proof. $\quad\square$

### B.3   OMITTED DETAILS IN SECTION 7.3 (BOUNDING GR ERROR)

*Proof of Lemma 6.* In this proof, we first show an upper bound on $\mathbb{E}\left[|\mu'_{t,i} - \widehat{\ell}_{t,i}||\widehat{\ell}_1, \ldots, \widehat{\ell}_{t-1}\right]$.

In the beginning, we rewrite the loss estimate as $\widehat{\ell}_{t,i} = \mathbb{I}\{a_t = i\}\mathbb{I}\{|\ell_{t,i}| \leqslant r\}M_t\ell_{t,i}$. And then, its conditional expectation (given all the loss estimates so far) can be rewritten as

$$
\begin{aligned}
\mathbb{E}[\widehat{\ell}_{t,i}|\widehat{\ell}_1,\ldots,\widehat{\ell}_{t-1}] &= \mathbb{E}[\mathbb{I}\{a_t = i\}\mathbb{I}\{|\ell_{t,i}| \leqslant r\}M_t\ell_{t,i}|\widehat{\ell}_1,\ldots,\widehat{\ell}_{t-1}] \\
&\overset{(a)}{=} \sum_{j=1}^K w_{t,j}\mathbb{E}[\mathbb{I}\{a_t = i\}\mathbb{I}\{|\ell_{t,i}| \leqslant r\}M_t\ell_{t,i}|\widehat{\ell}_1,\ldots,\widehat{\ell}_{t-1},a_t = j] \\
&\overset{(b)}{=} w_{t,i}\mathbb{E}[\mathbb{I}\{|\ell_{t,i}| \leqslant r\}M_t\ell_{t,i}|\widehat{\ell}_1,\ldots,\widehat{\ell}_{t-1},a_t = i] \\
&\overset{(c)}{=} w_{t,i}\mathbb{E}[M_t|\widehat{\ell}_1,\ldots,\widehat{\ell}_{t-1},a_t = i] \cdot \mathbb{E}[\mathbb{I}\{|\ell_{t,i}| \leqslant r\}\ell_{t,i}|\widehat{\ell}_1,\ldots,\widehat{\ell}_{t-1},a_t = i] \\
&= w_{t,i}\mathbb{E}[M_t|\widehat{\ell}_1,\ldots,\widehat{\ell}_{t-1},a_t = i] \cdot \mathop{\mathbb{E}}_{\ell_{t,i}\sim P_{t,i}}[\mathbb{I}\{|\ell_{t,i}| \leqslant r\}\ell_{t,i}],
\end{aligned} \tag{2}
$$

where (a) is from the law of total probability, (b) is based on the fact that only the summand with $j = i$ is non-zero, and (c) is due to the fact that loss generation is independent of the GR process.

Taking the difference between $\mu'_{t,i}$ and $\mathbb{E}[\widehat{\ell}_{t,i}|\widehat{\ell}_1,\ldots,\widehat{\ell}_{t-1}]$, we get

$$
\begin{aligned}
&\mu'_{t,i} - \mathbb{E}\left[\widehat{\ell}_{t,i}|\widehat{\ell}_1,\ldots,\widehat{\ell}_{t-1}\right] \\
&= \mathbb{E}[\ell_{t,i} \cdot \mathbb{I}\{|\ell_{t,i}| \leqslant r\}] - \mathbb{E}[\widehat{\ell}_{t,i}|\widehat{\ell}_1,\ldots,\widehat{\ell}_{t-1}] \\
&\overset{(a)}{=} \left(1 - w_{t,i}\mathbb{E}[M_t|\widehat{\ell}_1,\ldots,\widehat{\ell}_{t-1},a_t = i]\right) \cdot \mathop{\mathbb{E}}_{\ell_{t,i}\sim P_{t,i}}[\mathbb{I}\{|\ell_{t,i}| \leqslant r\}\ell_{t,i}] \\
&\overset{(b)}{=} (1 - w_{t,i})^{L_t} \mathop{\mathbb{E}}_{\ell_{t,i}\sim P_{t,i}}[\mathbb{I}\{|\ell_{t,i}| \leqslant r\}|\ell_{t,i}|] \\
&\leqslant (1 - w_{t,i})^{L_t} \mathop{\mathbb{E}}_{\ell_{t,i}\sim P_{t,i}}[|\ell_{t,i}|] \\
&\overset{(c)}{\leqslant} (1 - w_{t,i})^{L_t} \mathop{\mathbb{E}}_{\ell_{t,i}\sim P_{t,i}}[|\ell_{t,i}|^\alpha]^{1/\alpha} \\
&\leqslant \sigma(1 - w_{t,i})^{L_t},
\end{aligned} \tag{3}
$$

where (a) is from Eq. (2), (b) is from Lemma 11, and (c) follows from Hölder's inequality.

Similarly, we have

$$
\mathbb{E}\left[\widehat{\ell}_{t,i}|\widehat{\ell}_1,\ldots,\widehat{\ell}_{t-1}\right] - \mu'_{t,i} \leqslant \sigma(1 - w_{t,i})^{L_t}. \tag{4}
$$

Combining Eqs. (3) and (4), we get

$$
\mathbb{E}\left[|\mu'_{t,i} - \widehat{\ell}_{t,i}||\widehat{\ell}_1,\ldots,\widehat{\ell}_{t-1}\right] \leqslant \sigma(1 - w_{t,i})^{L_t}. \tag{5}
$$

Now, we decompose the GR Error into two terms by

$$
\mathrm{GRERR} = \underbrace{\mathbb{E}\left[\sum_{t=1}^T \langle w_t, \mu'_t - \widehat{\ell}_t\rangle\right]}_{\mathrm{GRERR\ I}} + \underbrace{\mathbb{E}\left[\sum_{t=1}^T \langle e_{i^*}, \widehat{\ell}_t - \mu'_t\rangle\right]}_{\mathrm{GRERR\ II}}.
$$

and bound these two terms separately.

**Bounding GRERR I.** To bound GRERR I, we have

$$
\begin{aligned}
\text{GRERR I} &= \sum_{t=1}^{T} \sum_{i=1}^{K} \mathbb{E}\left[ w_{t,i}(\mu'_{t,i} - \widehat{\ell}_{t,i}) \right] \\
&= \sum_{t=1}^{T} \sum_{i=1}^{K} \mathbb{E}\left[ \mathbb{E}\left[ w_{t,i}(\mu'_{t,i} - \widehat{\ell}_{t,i}) | \widehat{\ell}_1, \ldots, \widehat{\ell}_{t-1} \right] \right] \\
&= \sum_{t=1}^{T} \sum_{i=1}^{K} \mathbb{E}\left[ w_{t,i}\mathbb{E}\left[ (\mu'_{t,i} - \widehat{\ell}_{t,i}) | \widehat{\ell}_1, \ldots, \widehat{\ell}_{t-1} \right] \right] \\
&\overset{(a)}{\leqslant} \sum_{t=1}^{T} \sum_{i=1}^{K} \mathbb{E}[\sigma \cdot w_{t,i}(1 - w_{t,i})^{L_t}] \\
&\overset{(b)}{\leqslant} \frac{\sigma K}{e} \cdot \mathbb{E}\left[ \sum_{t=1}^{T} \frac{1}{L_t} \right],
\end{aligned}
$$

where (a) is from Eq. (5), and (b) is from the elementary inequality $x(1-x)^{L_t} \leqslant xe^{-xL_t} \leqslant 1/(eL_t), \forall x \in [0,1]$, given any $L_t > 0$ (Neu & Bartók, 2016, Lemma 5).

Therefore, by choosing $L_t \geqslant K^{\frac{1}{\alpha}}T^{1-\frac{1}{\alpha}}/e$, we ensure that

$$
\text{GRERR I} \leqslant \sigma K^{1-\frac{1}{\alpha}}T^{\frac{1}{\alpha}}.
$$

**Bounding GRERR II.** To begin with, we bound GRERR II by

$$
\begin{aligned}
\text{GRERR II} &= \sum_{t=1}^{T} \mathbb{E}\left[ \widehat{\ell}_{t,i^*} - \mu'_{t,i^*} \right] \\
&= \sum_{t=1}^{T} \mathbb{E}\left[ \mathbb{E}\left[ \widehat{\ell}_{t,i^*} - \mu'_{t,i^*} | \widehat{\ell}_1, \ldots, \widehat{\ell}_{t-1} \right] \right] \\
&\overset{(a)}{\leqslant} \sigma \sum_{t=1}^{T} \mathbb{E}\left[ (1 - w_{t,i^*})^{L_t} \right],
\end{aligned}
\tag{6}
$$

where step (a) is due to Eq. (5).

One may notice that, now the challenge is that we do not have a nice control of form $x(1-x)^{L_t}$ as in the first term (which was nicely bounded regardless of $x \in [0,1]$). Therefore, we instead obtain a lower bound on $w_{t,i^*}$, using the single-step stability lemma (Lemma 4).

Specifically, recall that $w_1 = (1/K, \ldots, 1/K)$ (due to the decision rule in Line 5). Combining it with Lemma 4, we have

$$
w_{t,i} \geqslant \frac{1}{K} \cdot \exp\left( -\eta \sum_{t'=1}^{t-1} \left\| \widehat{\ell}_{t'} \right\|_1 \right), \forall t \in [T], i \in [K], \eta > 0.
$$

Therefore, in order to ensure

$$
\sigma \sum_{t=1}^{T} \mathbb{E}\left[ (1 - w_{t,i^*})^{L_t} \right] \leqslant \sigma K^{1-\frac{1}{\alpha}}T^{\frac{1}{\alpha}},
$$

it suffices to choose $L_t$ such that

$$
\left( 1 - \frac{\exp\left( -\eta \sum_{t'=1}^{t-1} \left\| \widehat{\ell}_{t'} \right\|_1 \right)}{K} \right)^{L_t} \leqslant \left( \frac{K}{T} \right)^{1-\frac{1}{\alpha}},
$$

which yields

$$L_t \geqslant \left(1 - \frac{1}{\alpha}\right) \frac{\ln\left(\frac{K}{T}\right)}{\ln\left(1 - \frac{\exp\left(-\eta \sum_{t'=1}^{t-1} \|\widehat{\ell}_{t'}\|_1\right)}{K}\right)}.$$

**Bounding Two Terms.** Combining two cases, choosing

$$L_t = \max\left\{\left(1 - \frac{1}{\alpha}\right) \frac{\ln\left(\frac{K}{T}\right)}{\ln\left(1 - \frac{\exp\left(-\eta \sum_{t'=1}^{t-1} \|\widehat{\ell}_{t'}\|_1\right)}{K}\right)}, K^{\frac{1}{\alpha}} T^{1-\frac{1}{\alpha}} / e\right\}$$

ensures that

$$\text{GRERR} \leqslant 2\sigma K^{1-\frac{1}{\alpha}} T^{\frac{1}{\alpha}}.$$

$\square$

*Remark* 13. As long as $r, L_1, \ldots, L_{t-1}$ are all finite, so is $L_t$. The reason is that $\left\|\widehat{\ell}_t\right\|_1 \leqslant rL_t, \forall t$, almost surely.

### B.4 UPPER BOUND AND LOWER BOUND FOR HEAVY-TAILED ADVERSARIAL BANDITS

#### B.4.1 PROOF OF THEOREM 1 (UPPER BOUND)

*Proof of Theorem 1.* Putting Lemmas 5, 6, 9, and 10 together, along with the choice of $\eta$, $r$, and $L_t$, we have,

$$\begin{aligned}
R_T &\leqslant 2T\sigma^\alpha r^{1-\alpha} + \frac{6\ln K}{\eta} + 2\eta\sigma^\alpha r^{2-\alpha}KT + 2\sigma K^{1-\frac{1}{\alpha}} T^{\frac{1}{\alpha}} \\
&= 2T\sigma^\alpha (\sigma T^{\frac{1}{\alpha}} K^{-\frac{1}{\alpha}})^{1-\alpha} + \frac{6}{\sigma^{-1} K^{\frac{1}{\alpha}-1} T^{-\frac{1}{\alpha}} \sqrt{\ln K}} \\
&\quad + 2\sigma^{-1} K^{\frac{1}{\alpha}-1} T^{-\frac{1}{\alpha}} \sqrt{\ln K} \sigma^\alpha (\sigma T^{\frac{1}{\alpha}} K^{-\frac{1}{\alpha}})^{2-\alpha} KT + 2\sigma K^{1-\frac{1}{\alpha}} T^{\frac{1}{\alpha}} \\
&= O(\sigma K^{1-\frac{1}{\alpha}} T^{\frac{1}{\alpha}} \sqrt{\ln K}).
\end{aligned}$$

$\square$

#### B.4.2 FORMAL STATEMENT AND PROOF OF LOWER BOUND

**Theorem 2.** *Consider the heavy-tailed MAB problem defined in Section 3. For any bandit algorithm, there must exist at one problem instance such that the algorithm suffers regret*

$$R_T = \Omega\left(\sigma K^{1-\frac{1}{\alpha}} T^{\frac{1}{\alpha}}\right).$$

*Proof of Theorem 2.* This proof is a direct modification based on Wu et al. (2023, Appendix B). We use $\pi$ to denote a bandit algorithm. We construct two environments, denoted by $\nu_1$ and $\nu_2$, respectively. And then we show that any algorithm suffers the claimed regret in one of these two environments.

**Environment $\nu_1$.** In $\nu_1$, the loss of action 1 in every round $t \in [T]$ is given by

$$\ell_1 = \begin{cases} \sigma/\gamma & \text{,with probability } \frac{1}{2}\gamma^\alpha \\ 0 & \text{,with probability } 1 - \frac{1}{2}\gamma^\alpha, \end{cases}$$

where $\gamma \leqslant 1$ is some free parameter to choose at the last step of the proof. One can verify that $\mathbb{E}[|\ell_1|^\alpha] \leqslant \sigma^\alpha$ via direct calculations.

For any suboptimal action $i \neq 1$, the loss in every round $t \in [T]$ is given by

$$\ell_i = \begin{cases} \sigma/\gamma & \text{,with probability } \frac{3}{10}\gamma^\alpha \\ 0 & \text{,with probability } 1 - \frac{3}{10}\gamma^\alpha. \end{cases}$$

One can verify that $\mathbb{E}[|\ell_i|^\alpha] \leqslant \sigma^\alpha, \forall i \neq 1$. Moreover, action 1 is the optimal one and we have the "sub-optimality gap" $\Delta := \mathbb{E}[\ell_1 - \ell_i] = \frac{\sigma}{5}\gamma^{\alpha-1}, \forall i \neq 1$.

For any $i \in [K]$, we use $N_i(T)$ to denote how many times action $i$ is taken in total by the end of round $T$ (which is a random variable). Given algorithm $\pi$ and environment $\nu_1$, we define $i' = \underset{a\in\{2,\dots,K\}}{\operatorname{argmin}} \underset{\pi,\nu_1}{\mathbb{E}}[N_i(T)]$, and hence we have $\underset{\pi,\nu_1}{\mathbb{E}}[N_{i'}(T)] \leqslant \frac{T}{K-1}$. Now we are able to construct the second environment.

**Environment $\nu_2$.** In this environment, the loss for action $i$ (denoted by $\ell'_i$) is the same as $\ell_i$, except that for action $i'$, now the loss follows

$$
\ell'_{i'} = \begin{cases} \sigma/\gamma & \text{,with probability } \dfrac{7}{10}\gamma^\alpha \\ 0 & \text{,with probability } 1 - \dfrac{7}{10}\gamma^\alpha. \end{cases}
$$

One can verify that $\mathbb{E}[\ell'_{i'}] = \frac{7}{10}\gamma^\alpha, \mathbb{E}[|\ell'_{i'}|^\alpha] \leqslant \sigma^\alpha$, and now the optimal action is $i'$. Moreover, $\mathbb{E}[\ell'_{i'} - \ell'_1] = \Delta$ and $\mathbb{E}[\ell'_{i'} - \ell'_i] = 2\Delta, \forall i \neq 1, i'$.

Now we are ready to follow the steps in Lattimore & Szepesvári (2020, Section 15.2) to obtain the lower bound. We use $R_T(\pi, \nu_1)$ and $R_T(\pi, \nu_2)$ to denote the regret incurred by algorithm $\pi$ interacting with environments $\nu_1$ and $\nu_2$ respectively and use $\mathbb{P}_{\pi,\nu_1}(\cdot), \mathbb{P}_{\pi,\nu_2}(\cdot)$ to denote the probability of an event yielded by algorithm $\pi$ interacting with $\nu_1$ and $\nu_2$ respectively. For two distributions $Q, Q'$ on the same space, their KL-divergence is denoted by $\mathrm{KL}(Q\|Q')$, and their Total Variation (TV) distance is denoted by $\mathrm{TV}(Q\|Q') := \underset{A \text{ measurable}}{\sup} |Q(A) - Q'(A)|$.

Following from the regret definition, we first have

$$
R_T(\pi, \nu_1) = \Delta\left(T - \underset{\pi,\nu_1}{\mathbb{E}}[N_1(T)]\right) \geqslant \frac{\Delta T}{2}\mathbb{P}_{\pi,\nu_1}\left(N_1(T) \leqslant \frac{T}{2}\right),
$$

$$
R_T(\pi, \nu_2) = \Delta\underset{\pi,\nu_2}{\mathbb{E}}[N_1(T)] + \sum_{i \notin \{1,i'\}} 2\Delta\underset{\pi,\nu_2}{\mathbb{E}}[N_i(T)] \geqslant \frac{\Delta T}{2}\mathbb{P}_{\pi,\nu_2}\left(N_1(T) \geqslant \frac{T}{2}\right).
$$

By adding them together, we have

$$
R_T(\pi,\nu_1) + R_T(\pi,\nu_2) \geqslant \frac{\Delta T}{2}\left(\mathbb{P}_{\pi,\nu_1}\left(N_1(T) \leqslant \frac{T}{2}\right) + \mathbb{P}_{\pi,\nu_2}\left(N_1(T) \geqslant \frac{T}{2}\right)\right)
$$

$$
\overset{\text{(a)}}{\geqslant} \frac{\Delta T}{4}\exp\left(-\mathrm{KL}(\mathbb{P}_{\pi,\nu_1}\|\mathbb{P}_{\pi,\nu_2})\right),
$$

where step (a) follows from the Bretagnolle–Huber inequality (Lattimore & Szepesvári, 2020, Theorem 14.2).

It is left to bound $\mathrm{KL}(\mathbb{P}_{\pi,\nu_1}\|\mathbb{P}_{\pi,\nu_2})$. We abuse the notation a little bit and use $\ell_{t,i}$ or $\ell'_{t,i}$ to denote the corresponding loss distribution. Lattimore & Szepesvári (2020, Lemma 15.1) yields

$$
\mathrm{KL}(\mathbb{P}_{\pi,\nu_1}\|\mathbb{P}_{\pi,\nu_2}) = \underset{\pi,\nu_1}{\mathbb{E}}[N_1(T)] \cdot \mathrm{KL}(\ell_{t,i'}\|\ell'_{t,i'})
$$

$$
= \underset{\pi,\nu_1}{\mathbb{E}}[N_1(T)] \cdot \mathrm{KL}\left(\mathrm{Ber}(\frac{3}{10}\gamma^\alpha)\|\mathrm{Ber}(\frac{7}{10}\gamma^\alpha)\right)
$$

$$
\leqslant \underset{\pi,\nu_1}{\mathbb{E}}[N_1(T)] \cdot \frac{(\frac{3}{10}\gamma^\alpha - \frac{7}{10}\gamma^\alpha)^2}{\frac{7}{10}\gamma^\alpha(1 - \frac{7}{10}\gamma^\alpha)}
$$

$$
= \underset{\pi,\nu_1}{\mathbb{E}}[N_1(T)] \cdot \frac{\frac{8}{35}\gamma^\alpha}{(1 - \frac{7}{10}\gamma^\alpha)}
$$

$$
\leqslant \frac{4}{5}\underset{\pi,\nu_1}{\mathbb{E}}[N_1(T)] \cdot \gamma^\alpha,
$$

where the last step is simply because $\gamma^\alpha \leqslant 1$ and hence $1 - \frac{7}{10}\gamma^\alpha \geqslant \frac{3}{10}$.

Therefore, we have

$$R_T(\pi, \nu_1) + R_T(\pi, \nu_2) \geqslant \frac{\sigma \gamma^{\alpha-1} T}{20} \exp\left(-\frac{4T\gamma^\alpha}{5(K-1)}\right).$$

Choosing $\gamma = (\frac{K-1}{T})^{\frac{1}{\alpha}}$ (which is no larger than 1 since $K \leqslant T$ and $\alpha > 1$), we have

$$\max\{R_T(\pi, \nu_1), R_T(\pi, \nu_2)\} \geqslant \Omega(\sigma K^{1-\frac{1}{\alpha}} T^{\frac{1}{\alpha}}),$$

which completes the proof. $\qquad\square$

## C OMITTED DETAILS IN SECTION 8

### C.1 RESTATEMENT AND PROOF OF LEMMA 7

**Lemma 7.** Running Algorithm 1 modified skipping threshold $r' = \min\{\frac{1}{\eta K}, \frac{\sigma}{(2\beta)^{\frac{1}{\alpha}}}, \left(\frac{2\sigma^\alpha}{\eta\beta K}\right)^{\frac{1}{1+\alpha}}\}$ and Laplace distribution parameter $\eta' = \min\{\frac{\beta^{\frac{1}{\alpha}}}{\sigma K}, \left(\frac{3}{2}\right)^{\frac{1}{\alpha}} \frac{(\ln K)^{\frac{1}{\alpha}}}{\sigma T^{\frac{1}{\alpha}} K^{1-\frac{1}{\alpha}}}\}$ ensures

$$R_T = O\left(\sigma(T^{\frac{1}{\alpha}} K^{1-\frac{1}{\alpha}} (\ln K)^{1-\frac{1}{\alpha}} + T\beta^{1-\frac{1}{\alpha}})\right).$$

*Remark* 14. In this proof we will ignore the GR Error, since as long as we know of what order we want to control it, we can follow the proof of Lemma 6 to determine sufficiently large $L_t$ accordingly.

*Proof of Lemma 7.* We first decompose the regret by

$$R_T = \underbrace{\mathbb{E}\left[\sum_{t=1}^T \langle w_t - e_{i^*}, \mu_t - \mu_t'' \rangle\right]}_{\text{SKIPERR}'} + \underbrace{\mathbb{E}\left[\sum_{t=1}^T \langle w_t - e_{i^*}, \widehat{\ell}_t \rangle\right]}_{\text{FTPLREG}'},$$

where $\mu_t'' := (\mu_{t,1}'', \dots, \mu_{t,K}'')$ and $\mu_{t,i}'' := \underset{\ell \sim P_{\beta,t,i}}{\mathbb{E}}[\ell \cdot \mathbb{I}\{|\ell| \leqslant r\}], \forall i \in [K], t \in [T]$ is the "contaminated skipped mean", and we are going to bound these two terms separately.

**Bounding SKIPERR'.** We start with bounding $|\mu_{t,i} - \mu_{t,i}''|$. In particular, we have

$$\mu_{t,i} - \mu_{t,i}'' = \underset{\ell \sim P_{t,i}}{\mathbb{E}}[\ell] - \underset{\ell \sim P_{\beta,t,i}}{\mathbb{E}}[\ell \cdot \mathbb{I}\{|\ell| \leqslant r\}]$$

$$= \left(\underset{\ell \sim P_{t,i}}{\mathbb{E}}[\ell] - \underset{\ell \sim P_{t,i}}{\mathbb{E}}[\ell \cdot \mathbb{I}\{|\ell| \leqslant r\}]\right) + \left(\underset{\ell \sim P_{t,i}}{\mathbb{E}}[\ell \cdot \mathbb{I}\{|\ell| \leqslant r\}] - \underset{\ell \sim P_{\beta,t,i}}{\mathbb{E}}[\ell \cdot \mathbb{I}\{|\ell| \leqslant r\}]\right)$$

$$= \underset{\ell \sim P_{t,i}}{\mathbb{E}}[\ell \cdot \mathbb{I}\{|\ell| > r\}] + \left(\underset{\ell \sim P_{t,i}}{\mathbb{E}}[\ell \cdot \mathbb{I}\{|\ell| \leqslant r\}] - \underset{\ell \sim P_{\beta,t,i}}{\mathbb{E}}[\ell \cdot \mathbb{I}\{|\ell| \leqslant r\}]\right)$$

$$\leqslant \sigma^\alpha r^{1-\alpha} + \left(\underset{\ell \sim P_{t,i}}{\mathbb{E}}[\ell \cdot \mathbb{I}\{|\ell| \leqslant r\}] - \underset{\ell \sim P_{\beta,t,i}}{\mathbb{E}}[\ell \cdot \mathbb{I}\{|\ell| \leqslant r\}]\right)$$

$$= \sigma^\alpha r^{1-\alpha} + \beta \underbrace{\left(\underset{\ell \sim P_{t,i}}{\mathbb{E}}[\ell \cdot \mathbb{I}\{|\ell| \leqslant r\}] - \underset{\ell \sim Q_{t,i}}{\mathbb{E}}[\ell \cdot \mathbb{I}\{|\ell| \leqslant r\}]\right)}_{\leqslant 2r}$$

$$+ (1-\beta) \underbrace{\left(\underset{\ell \sim P_{t,i}}{\mathbb{E}}[\ell \cdot \mathbb{I}\{|\ell| \leqslant r\}] - \underset{\ell \sim P_{t,i}}{\mathbb{E}}[\ell \cdot \mathbb{I}\{|\ell| \leqslant r\}]\right)}_{=0}$$

$$\leqslant \sigma^\alpha r^{1-\alpha} + 2\beta r.$$

Similarly, we have

$$
\begin{aligned}
\mu''_{t,i} - \mu_{t,i} &= \mathop{\mathbb{E}}_{\ell \sim P_{\beta,t,i}} [\ell \cdot \mathbb{I}\{|\ell| \leqslant r\}] - \mathop{\mathbb{E}}_{\ell \sim P_{t,i}} [\ell] \\
&= \left( \mathop{\mathbb{E}}_{\ell \sim P_{t,i}} [\ell \cdot \mathbb{I}\{|\ell| \leqslant r\}] - \mathop{\mathbb{E}}_{\ell \sim P_{t,i}} [\ell] \right) + \left( \mathop{\mathbb{E}}_{\ell \sim P_{\beta,t,i}} [\ell \cdot \mathbb{I}\{|\ell| \leqslant r\}] - \mathop{\mathbb{E}}_{\ell \sim P_{t,i}} [\ell \cdot \mathbb{I}\{|\ell| \leqslant r\}] \right) \\
&\leqslant \mathop{\mathbb{E}}_{\ell \sim P_{t,i}} [|\ell| \cdot \mathbb{I}\{|\ell| > r\}] + \left( \mathop{\mathbb{E}}_{\ell \sim P_{\beta,t,i}} [\ell \cdot \mathbb{I}\{|\ell| \leqslant r\}] - \mathop{\mathbb{E}}_{\ell \sim P_{t,i}} [\ell \cdot \mathbb{I}\{|\ell| \leqslant r\}] \right) \\
&\leqslant \sigma^\alpha r^{1-\alpha} + \left( \mathop{\mathbb{E}}_{\ell \sim P_{\beta,t,i}} [\ell \cdot \mathbb{I}\{|\ell| \leqslant r\}] - \mathop{\mathbb{E}}_{\ell \sim P_{t,i}} [\ell \cdot \mathbb{I}\{|\ell| \leqslant r\}] \right) \\
&= \sigma^\alpha r^{1-\alpha} + \beta \underbrace{\left( \mathop{\mathbb{E}}_{\ell \sim Q_{t,i}} [\ell \cdot \mathbb{I}\{|\ell| \leqslant r\}] - \mathop{\mathbb{E}}_{\ell \sim P_{t,i}} [\ell \cdot \mathbb{I}\{|\ell| \leqslant r\}] \right)}_{\leqslant 2r} \\
&\quad + (1-\beta) \underbrace{\left( \mathop{\mathbb{E}}_{\ell \sim P_{t,i}} [\ell \cdot \mathbb{I}\{|\ell| \leqslant r\}] - \mathop{\mathbb{E}}_{\ell \sim P_{t,i}} [\ell \cdot \mathbb{I}\{|\ell| \leqslant r\}] \right)}_{=0} \\
&\leqslant \sigma^\alpha r^{1-\alpha} + 2\beta r.
\end{aligned}
$$

Therefore, we have

$$
|\mu''_{t,i} - \mu_{t,i}| \leqslant \sigma^\alpha r^{1-\alpha} + 2\beta r
$$

and hence we arrive at

$$
\textsc{SkipErr}' \leqslant 2T \left( \sigma^\alpha r^{1-\alpha} + 2\beta r \right)
$$

as in the proof of Lemma 9.

**Bounding FTPLREG'.** The proof for this part deviates that of Lemma 5 starting from its last step. Specifically, now we have

$$
\begin{aligned}
\mathbb{E}\left[ \sum_{i=1}^K (w_{t,i} - w_{t+1,i}) \widehat{\ell}_{t,i} | \widehat{\ell}_1, \dots, \widehat{\ell}_{t-1} \right] &\leqslant \eta \sum_{i=1}^K w_{t,i} \mathbb{E}[\mathbb{I}\{a_t = i\} (M_t)^2 (\tilde{\ell}_{t,i})^\alpha r^{2-\alpha} | \widehat{\ell}_1, \dots, \widehat{\ell}_{t-1}] \\
&\leqslant 2\eta \left( (1-\beta)\sigma^\alpha r^{2-\alpha} + \beta r^2 \right) \sum_{i=1}^K w_{t,i} \mathbb{E}[\mathbb{I}\{a_t = i\} (M_t)^2 | \widehat{\ell}_1, \dots, \widehat{\ell}_{t-1}] \\
&\leqslant 2\eta \left( (1-\beta)\sigma^\alpha r^{2-\alpha} + \beta r^2 \right) K,
\end{aligned}
$$

and therefore the entire FTPL Regret is bounded by

$$
\textsc{FTPLReg}' \leqslant 2\eta \left( \sigma^\alpha r^{2-\alpha} + \beta r^2 \right) KT + \frac{6 \ln K}{\eta}.
$$

**Choosing $\eta'$ and $r'$.** It is left to show that our choice of $\eta'$ and $r'$ yields the claimed upper bound. Combining the bounds on two terms shown above, we have

$$
R_T \leqslant \underbrace{2\sigma^\alpha r^{1-\alpha} T}_{\mathcal{T}_1(r)} + \underbrace{2\eta\sigma^\alpha r^{2-\alpha} KT}_{\mathcal{T}_2(r)} + \underbrace{4\beta r T}_{\mathcal{T}_3(r)} + \underbrace{2\eta\beta r^2 KT}_{\mathcal{T}_4(r)} + \frac{6 \ln K}{\eta}.
$$

We first determine $r'$ while treating $\eta$ as a given constant and then decide $\eta$.

To determine $r'$, we rely on some values $r_1$, $r_2$, and $r_3$ given by the following:

$$
\mathcal{T}_1(r) = \mathcal{T}_2(r) \to r_1 = \frac{1}{\eta K},
$$

$$
\mathcal{T}_1(r) = \mathcal{T}_3(r) \to r_2 = \frac{\sigma}{(2\beta)^{\frac{1}{\alpha}}},
$$

$$
\mathcal{T}_1(r) = \mathcal{T}_4(r) \to r_3 = \left( \frac{\sigma^\alpha}{\eta\beta K} \right)^{\frac{1}{1+\alpha}}.
$$

By choosing $r' = \min\{r_1, r_2, r_3\}$, we have

$$R_T \leqslant 2\mathcal{T}_1(r_1) + 2\mathcal{T}_1(r_2) + 2\mathcal{T}_1(r_3) + \frac{6\ln K}{\eta}$$

$$= 4\sigma^\alpha \left(\frac{1}{\eta K}\right)^{1-\alpha} T + 4\sigma^\alpha \left(\frac{\sigma}{(2\beta)^{\frac{1}{\alpha}}}\right)^{1-\alpha} T + 4\sigma^\alpha \left(\left(\frac{\sigma^\alpha}{\eta\beta K}\right)^{\frac{1}{1+\alpha}}\right)^{1-\alpha} T + \frac{6\ln K}{\eta}$$

$$\leqslant \underbrace{4\sigma^\alpha(\eta K)^{\alpha-1}T}_{\mathcal{T}_1'(\eta)} + 4\sigma\beta^{1-\frac{1}{\alpha}}T + \underbrace{4\sigma^{\frac{2\alpha}{\alpha+1}}(\eta\beta K)^{\frac{\alpha-1}{\alpha+1}}T}_{\mathcal{T}_2'(\eta)} + \underbrace{\frac{6\ln K}{\eta}}_{\mathcal{T}_3'(\eta)}.$$

To see the first inequality of this step, suppose $r' = \min\{r_1, r_2, r_3\} = r_1$, then

$$\mathcal{T}_1(r') + \mathcal{T}_2(r') + \mathcal{T}_3(r') + \mathcal{T}_4(r') = \mathcal{T}_1(r_1) + \mathcal{T}_2(r_1) + \mathcal{T}_3(r_1) + \mathcal{T}_4(r_1)$$
$$= 2\mathcal{T}_1(r_1) + \mathcal{T}_3(r_1) + \mathcal{T}_4(r_1)$$
$$\leqslant 2\mathcal{T}_1(r_1) + \mathcal{T}_3(r_2) + \mathcal{T}_4(r_3)$$
$$= 2\mathcal{T}_1(r_1) + \mathcal{T}_1(r_2) + \mathcal{T}_1(r_3),$$

where the inequality is because both $\mathcal{T}_3$ and $\mathcal{T}_4$ are monotonically increasing w.r.t. the input.

Then, suppose $r' = \min\{r_1, r_2, r_3\} = r_2$, we have

$$\mathcal{T}_1(r') + \mathcal{T}_2(r') + \mathcal{T}_3(r') + \mathcal{T}_4(r') = \mathcal{T}_1(r_2) + \mathcal{T}_2(r_2) + \mathcal{T}_3(r_2) + \mathcal{T}_4(r_2)$$
$$= \mathcal{T}_2(r_2) + 2\mathcal{T}_1(r_2) + \mathcal{T}_4(r_2)$$
$$\leqslant \mathcal{T}_2(r_1) + 2\mathcal{T}_1(r_2) + \mathcal{T}_4(r_3)$$
$$= \mathcal{T}_1(r_1) + 2\mathcal{T}_1(r_2) + \mathcal{T}_1(r_3).$$

Similarly, one can show that when $r' = \min\{r_1, r_2, r_3\} = r_3$, we have

$$\mathcal{T}_1(r') + \mathcal{T}_2(r') + \mathcal{T}_3(r') + \mathcal{T}_4(r') = \mathcal{T}_1(r_3) + \mathcal{T}_2(r_3) + \mathcal{T}_3(r_3) + \mathcal{T}_4(r_3)$$
$$= \mathcal{T}_2(r_3) + \mathcal{T}_3(r_3) + 2\mathcal{T}_1(r_3)$$
$$\leqslant \mathcal{T}_2(r_1) + \mathcal{T}_3(r_2) + 2\mathcal{T}_1(r_3)$$
$$= \mathcal{T}_1(r_1) + \mathcal{T}_1(r_2) + 2\mathcal{T}_1(r_3).$$

Combining all these three cases yields

$$\mathcal{T}_1(r') + \mathcal{T}_2(r') + \mathcal{T}_3(r') + \mathcal{T}_4(r') \leqslant 2\mathcal{T}_1(r_1) + 2\mathcal{T}_1(r_2) + 2\mathcal{T}_1(r_3).$$

Now we determine $\eta'$ via first obtaining $\eta_1$ and $\eta_2$:

$$\mathcal{T}_1'(\eta) = \mathcal{T}_2'(\eta) \rightarrow \eta_1 = \frac{\beta^{\frac{1}{\alpha}}}{\sigma K},$$

$$\mathcal{T}_1'(\eta) = \mathcal{T}_3'(\eta) \rightarrow \eta_2 = \left(\frac{3}{2}\right)^{\frac{1}{\alpha}} \frac{(\ln K)^{\frac{1}{\alpha}}}{\sigma T^{\frac{1}{\alpha}} K^{1-\frac{1}{\alpha}}}.$$

By choosing $\eta' = \min\{\eta_1, \eta_2\}$, we have

$$R_T \leqslant 2\mathcal{T}_1'(\eta_1) + 2\mathcal{T}_1'(\eta_2) + 8\sigma\beta^{1-\frac{1}{\alpha}}T$$
$$= O\left(\sigma(T^{\frac{1}{\alpha}} K^{1-\frac{1}{\alpha}}(\ln K)^{1-\frac{1}{\alpha}} + T\beta^{1-\frac{1}{\alpha}})\right).$$

$\square$

## C.2 Formal Statement and Proof of Lower Bound for Heavy-tailed Adversarial Bandits with Huber Contamination

**Theorem 3** (Lower Bound for Heavy-tailed Adversarial Bandits with Huber Contamination). *For heavy-tailed adversarial bandits with Huber contamination defined in Definition 1, and for any bandit algorithm, there must exist one problem instance such that the algorithm suffers regret*

$$R_T = \Omega\left(\sigma\left(T^{\frac{1}{\alpha}} K^{1-\frac{1}{\alpha}} + T\beta^{1-\frac{1}{\alpha}}\right)\right)$$

*Proof of Theorem 3.* Theorem 2 indicates that, in the presence of heavy tails, every algorithm suffers $\Omega\left(\sigma T^{\frac{1}{\alpha}}K^{1-\frac{1}{\alpha}}\right)$ regret in the worst case, *regardless of contamination level $\beta$*. To see this, one can let $P_{t,i} = Q_{t,i}, \forall t \in [T], i \in [K]$ so that $P_{\beta,t,i} = P_{t,i}$ and there is equivalently no contamination.

Therefore, to show the lower bound with contamination, it suffices to show a lower bound of

$$\Omega\left(\sigma T \beta^{1-\frac{1}{\alpha}}\right)$$

given any $\beta \in (0,1]$.

To begin with, we construct two environments, denoted by $\nu_1'$ and $\nu_2'$, respectively, and $\nu_1'$ is exactly the same as $\nu_1$ given in the proof of Theorem 2. And then, we use $\tilde{\nu}_1'$ to denote the contaminated version of $\nu_1'$ (the bad distributions will be determined later).

Given algorithm $\pi$ and environment $\nu_1'$, we define $i'' = \underset{a\in\{2,...,K\}}{\operatorname{argmin}} \underset{\pi,\nu_1'}{\mathbb{E}}[N_i(T)]$, and hence we have $\underset{\pi,\nu_1'}{\mathbb{E}}[N_{i''}(T)] \leqslant \frac{T}{K-1}$. Now we are able to construct the second environment.

**Environment $\nu_2'$.** In this environment, everything is the same as in $\nu_1'$, except that for action $i''$, now the loss follows

$$\ell_{i''}' = \begin{cases} \sigma/\gamma & \text{,with probability } \frac{7}{10}\gamma^\alpha \\ 0 & \text{,with probability } 1 - \frac{7}{10}\gamma^\alpha. \end{cases}$$

One can verify that $\mathbb{E}[\ell_{i''}'] = \frac{7}{10}\gamma^\alpha$, $\mathbb{E}[|\ell_{i''}'|^\alpha] \leqslant \sigma^\alpha$ and now the optimal action is $i''$. We use $\tilde{\nu}_2'$ to denote the contaminated version of $\nu_2'$ (where the bad distributions are determined later).

We choose $\gamma = \beta^{\frac{1}{\alpha}} \in (0,1]$. Then for any $i \in [K]$, we have $\mathrm{TV}(\ell_i \| \ell_i') \leqslant \frac{2}{5}\gamma^\alpha = \frac{2}{5}\beta \leqslant \frac{\beta}{1-\beta}$. According to Lemma 13, for any action $i \in [K]$, there exist bad distributions $G_i$ and $G_i'$ such that

$$(1-\beta)\ell_i + \beta G_i = (1-\beta)\ell_i' + \beta G_i',$$

and we construct $\tilde{\nu}_1'$ and $\tilde{\nu}_2'$ by

$$\tilde{\nu}_1' = \{x_i = (1-\beta)\ell_i + \beta G_i : i \in [K]\},$$
$$\tilde{\nu}_2' = \{x_i' = (1-\beta)\ell_i' + \beta G_i' : i \in [K]\},$$

where $x_i$ and $x_i'$ denote the loss distributions for action $i$ in these two environments, respectively.

Following from the regret definition, we first have

$$R_T(\pi, \tilde{\nu}_1') = \Delta\left(T - \underset{\pi,\tilde{\nu}_1'}{\mathbb{E}}[N_1(T)]\right) \geqslant \frac{\Delta T}{2}\mathbb{P}_{\pi,\tilde{\nu}_1'}\left(N_1(T) \leqslant \frac{T}{2}\right),$$

$$R_T(\pi, \tilde{\nu}_2') = \Delta\underset{\pi,\tilde{\nu}_2'}{\mathbb{E}}[N_1(T)] + \sum_{i\notin\{1,i''\}} 2\Delta\underset{\pi,\tilde{\nu}_2'}{\mathbb{E}}[N_i(T)] \geqslant \frac{\Delta T}{2}\mathbb{P}_{\pi,\tilde{\nu}_2'}\left(N_1(T) \geqslant \frac{T}{2}\right).$$

By adding them together, we have

$$\begin{aligned} R_T(\pi, \tilde{\nu}_1') + R_T(\pi, \tilde{\nu}_2') &\geqslant \frac{\Delta T}{2}\left(\mathbb{P}_{\pi,\tilde{\nu}_1'}\left(N_1(T) \leqslant \frac{T}{2}\right) + \mathbb{P}_{\pi,\tilde{\nu}_2'}\left(N_1(T) \geqslant \frac{T}{2}\right)\right) \\ &\overset{(a)}{\geqslant} \frac{\Delta T}{4}\exp\left(-\mathrm{KL}(\mathbb{P}_{\pi,\tilde{\nu}_1'}\|\mathbb{P}_{\pi,\tilde{\nu}_2'})\right) \\ &\overset{(b)}{=} \frac{\Delta T}{4}\exp(0), \end{aligned}$$

where step (a) follows from the Bretagnolle–Huber inequality (Lattimore & Szepesvári, 2020, Theorem 14.2), and step (b) is due to the fact that $\tilde{\nu}_1'$ and $\tilde{\nu}_2'$ are identical under our construction.

Recall that $\Delta = \frac{\sigma}{5}\gamma^{\alpha-1}$ and $\gamma = \beta^{\frac{1}{\alpha}}$, we arrive at

$$\max\{R_T(\pi, \nu_1), R_T(\pi, \nu_2)\} \geqslant \Omega(\sigma T \beta^{1-\frac{1}{\alpha}}),$$

which completes the proof together with Theorem 2. $\qquad\square$

# D  OMITTED DETAILS IN SECTION 9

## D.1  PROOF OF COROLLARY 1

*Proof of Corollary 1.* To show the regret guarantee, it suffices to show that the observed loss always satisfies Assumption 1 with $\sigma = O(1/\varepsilon)$ and $\alpha = 2$.

Recall that now all losses are deterministic and lie in $[0, 1]$, let $b \in [0, 1]$ be the loss for some action in some round, then the corresponding observed loss $\tilde{b}$ has PDF $f'(x) := \varepsilon \cdot \exp(-\varepsilon|x - b|)/2, \forall x \in \mathbb{R}$ for some $\varepsilon \in (0, 1]$, and its second raw moment is equal to $2(1/\varepsilon)^2 + b^2 \leqslant 2(1/\varepsilon)^2 + 1 \leqslant 3(1/\varepsilon)^2 = (\sqrt{3}/\varepsilon)^2$. Therefore, running Algorithm 1 with $\sigma = \sqrt{3}/\varepsilon$ and $\alpha = 2$ ensures $R_T = O(\sqrt{KT \ln K}/\varepsilon)$. $\qquad\qquad\square$

# E  AUXILIARY LEMMAS

In this section, we provide some auxiliary lemmas used in this paper.

**Lemma 11** ((Part of) Lemma 4 of Neu & Bartók (2016)). *Consider the Geometric Resampling process defined in Algorithm 1, we have*

$$\mathbb{E}\left[M_t \big| \widehat{\ell}_1, \ldots, \widehat{\ell}_{t-1}, a_t = i\right] = \frac{1 - (1 - w_{t,i})^{L_t}}{w_{t,i}}.$$

*Proof.* By direct calculation, we have

$$\mathbb{E}\left[M_t \big| \widehat{\ell}_1, \ldots, \widehat{\ell}_{t-1}, a_t = i\right] = \sum_{n=1}^{\infty} n(1 - w_{t,i})^{n-1} w_{t,i} - \sum_{n=L_t}^{\infty} (n - L_t)(1 - w_{t,i})^{n-1} w_{t,i}$$

$$= \left(1 - (1 - w_{t,i})^{L_t}\right) \sum_{n=1}^{\infty} n(1 - w_{t,i})^{n-1} w_{t,i} = \frac{1 - (1 - w_{t,i})^{L_t}}{w_{t,i}}.$$

$\qquad\qquad\square$

**Lemma 12** ((Part of) Lemma 39 of Dai et al. (2022)). *Consider the Geometric Resampling process defined in Algorithm 1, we have*

$$\mathbb{E}\left[(M_t)^2 \big| \widehat{\ell}_1, \ldots, \widehat{\ell}_{t-1}, a_t = i\right] \leqslant \frac{2}{(w_{t,i})^2}.$$

*Proof.* Notice that $M_t$ is stochastically dominated by a Geometric distribution with parameter $w_{t,i}$ (denoted by $\text{Geo}(w_{t,i})$), its second moment is bounded by

$$\mathbb{E}\left[(M_t)^2 \big| \widehat{\ell}_1, \ldots, \widehat{\ell}_{t-1}, a_t = i\right] \leqslant \mathbb{E}[(\text{Geo}(w_{t,i}))^2] = \text{Var}(\text{Geo}(w_{t,i})) + \left(\mathbb{E}[\text{Geo}(w_{t,i})]\right)^2$$

$$= \frac{1 - w_{t,i}}{(w_{t,i})^2} + \frac{1}{(w_{t,i})^2} \leqslant \frac{2}{(w_{t,i})^2}.$$

$\qquad\qquad\square$

**Lemma 13** (Theroem 5.1 of Chen et al. (2018)). *Let $R_1$ and $R_2$ be two distributions on $\mathcal{X}$. If for some $\beta \in [0, 1]$ it holds that $\text{TV}(R_1 \| R_2) \leqslant \frac{\beta}{1-\beta}$, then there exist two distributions on the same probability space $G_1$ and $G_2$ such that*

$$(1 - \beta)R_1 + G_1 = (1 - \beta)R_2 + G_2.$$

# F  DERIVATION OF OMD WITH LOG-BARRIER FOR HEAVY-TAILED ADVERSARIAL BANDITS

In this section, we present how to adopt a data-dependent regret guarantee achieved by OMD with log-barrier (with specific choices of update rule, loss estimator, and learning rate $\eta'$ to be

determined later) from Wei & Luo (2018, Theorem 4) to achieve near-optimal regret bound of $O(\sigma K^{1-1/\alpha} T^{1/\alpha} (\ln T)^2)$. The derivations were suggested by the reviewer J99q.[9]

It suffices to show that, after skipping with $r = \sigma (T/K)^{1/\alpha}$ (so that true losses are bounded in $[-r, r]$), the $O(\sigma K^{1-1/\alpha} T^{1/\alpha} (\ln T)^2)$ regret bound is ensured. Now, by further scaling all losses by $1/r$, all losses are bounded in $[-1, 1]$, and we can directly apply Theorem 4 of Wei & Luo (2018) and get

$$
\mathbb{E}\left[\langle w_t - u_{i^*}, \ell_t / r \rangle\right] = O\left(\mathbb{E}\left[\frac{K \ln T}{\eta'} + \eta' \sum_{t=1}^{T} \left(\frac{\ell_{t,i^*}}{r} - \sum_{t'=1}^{T} \frac{\ell_{t,i^*}}{rT}\right)^2 + K (\ln T)^2\right]\right).
$$

By multiplying $r$ on both sides (and now the LHS becomes exactly the regret definition), we have

$$
\begin{aligned}
\mathbb{E}\left[\langle w_t - u_{i^*}, \ell_t \rangle\right] &= O\left(\mathbb{E}\left[\frac{rK \ln T}{\eta'} + r\eta' \sum_{t=1}^{T} \left(\frac{\ell_{t,i^*}}{r} - \sum_{t'=1}^{T} \left(\frac{\ell_{t,i^*}}{rT}\right)\right)^2 + rK (\ln T)^2\right]\right) \\
&\leqslant O\left(\mathbb{E}\left[\frac{rK \ln T}{\eta'} + r\eta' \sum_{t=1}^{T} \left(\frac{\ell_{t,i^*}}{r}\right)^2 + rK (\ln T)^2\right]\right) \\
&\leqslant O\left(\mathbb{E}\left[\frac{rK \ln T}{\eta'} + \frac{\eta'}{r} \sum_{t=1}^{T} (\ell_{t,i^*})^\alpha r^{2-\alpha} + rK (\ln T)^2\right]\right) \\
&\leqslant O\left(\frac{rK \ln T}{\eta'} + \frac{\eta'}{r} T \sigma^\alpha r^{2-\alpha} + rK (\ln T)^2\right).
\end{aligned}
$$

Finally, by plugging in $r = \sigma (T/K)^{1/\alpha}$ and choosing $\eta' = \Theta(1)$, we get the desired regret bound.

---

[9]See "Weakness" in https://openreview.net/forum?id=jeMZi2Z9xe&noteId=J2d8meZpEu.

