# OpenReview forum: "Follow-the-Perturbed-Leader for Adversarial Bandits: Heavy Tails, Robustness, and Privacy"
_ICLR.cc/2024/Conference — ICLR 2024 Conference Withdrawn Submission_

### Official Review · Reviewer_PzqF · 2023-10-29

**Soundness:** 4 excellent
**Presentation:** 3 good
**Contribution:** 3 good
**Rating:** 8
**Confidence:** 3

**Summary:**

This submission studies the adversarial K-armed bandit problem with heavy-tailed losses. Before the game begins, an oblivious adversary selects a sequence of $T$ heavy-tailed loss distributions for each arm, after which the learner pulls a sequence of $T$ arms, and only observes the loss realizations for the arms they decided to pull. The authors' main result is to show that a modified version of Follow-the-Perturbed-Leader (FPTL) achieves optimal regret with respect to the best arm in hindsight, up to logarithmic-in-K factors. At a high level, FTPL adds a random perturbation to the sequence of observed losses at each time-step, before taking the action which minimizes the sequence of perturbed losses. The authors’ algorithm (Algorithm 1) proceeds similarly, with the key difference being that they ’skip’ losses (i.e. do not factor them into future calculations) if they are larger than a given cutoff threshold. In addition to achieving near-optimal performance in the adversarial setting with unbounded losses, the authors apply their algorithm to two different settings: adversarial bandits with Huber contamination and local differential privacy. In the Huber contamination setting, the loss observed by the learner is not the true loss, but is instead generated from some arbitrary and unknown distribution. Under this setting, the authors show that their algorithm achieves optimal regret (up to logarithmic factors). Under differential privacy, the authors’ algorithm improves upon existing results by polylogarithmic factors.

**Strengths:**

The authors show that a slight modification of a well-known and popular algorithm (FTPL) achieves near-optimal performance in the adversarial multi-armed bandit setting with heavy-tailed losses. Their algorithm is applicable in a wider range of settings when compared to previous, Follow-the-Regularized-Leader (FTRL)-based algorithms.  (See Assumption 2 and the following discussion for more details.) Additionally, the two applications are interesting, and the authors’ retults in these settings are either near-optimal or state-of-the-art. Finally, the writing of the paper is overall a strength. While I had one question in particular (see below), I found that the authors did a good job of succinctly describing (1) their algorithm (2) the challenges of the setting they consider, (3) the salient parts of their analysis, and (4) their applications.

**Weaknesses:**

It would be good to give some intuition behind why the algorithm ’skips’ losses.

Claiming Lemma 1 as a key observation seems to be an overstatement of the authors’ results, as Lemma 1 appears in previous work. It would be good if the authors could clarify what exactly is their observation when compared to previous work.

**Questions:**

Can you results be extended to settings in which the loss distributions are generated  by an adaptive adversary?

Is the skipping of large losses necessary? Or could a more clever analysis remove the need for this step in the algorithm?

---

> ### Author Response · Authors · 2023-11-17
> **Response to Reviewer PzqF**
>
> We appreciate your time and your feedback.
>
> > **Comment 1:** It would be good to give some intuition behind why the algorithm `skips' losses. Is the skipping of large losses necessary? Or could a more clever analysis remove the need for this step in the algorithm?
>
> **Our response:** Thank you for the question. Note that the use of "skipping" (i.e., setting zero loss estimates whenever overly-large true losses are encountered under our FTPL framework) is to control the largest possible loss estimate (in terms of absolute value under $O(\sigma (T/K)^{1/\alpha})$) and avoid directly feeding the learning algorithm with unbounded loss estimates (to ensure stability). As one particular way to achieve this goal, "skipping" itself is not the key, and there could be other alternatives. For example, Dorn et al. (2023) used a similar "truncation" technique (i.e., projecting losses to a bounded range $[a,b]$ via $\text{proj}(x) := \text{argmin}_{x'\in [a,b]}|x-x'|$ rather than "skipping") in their FTRL-based design, and the minimax-optimal regret can also be achieved (in a simpler setting where heavy-tailed losses are non-negative). While we do not know whether "skipping" is necessary, we are not aware of any existing work that handles unbounded/negative losses in similar setups without explicit skipping (or truncation).
>
> > **Comment 2:** Claiming Lemma 1 as a key observation seems to be an overstatement of the authors’ results, as Lemma 1 appears in previous work. It would be good if the authors could clarify what exactly is their observation when compared to previous work.
>
> **Our response:** We agree that Lemma 1 is a technical result from previous work, and we would like to clarify that we did not mean to claim Lemma 1 as our finding. By "key observation", we meant that this existing stability property of FTPL could be leveraged to get around the issue of negative loss estimates suffered by FTRL (with regularizers, e.g., Tsallis entropy or Shannon entropy), as discussed in Section 5. We understand that this statement could be ambiguous and misleading. Therefore, we have improved the presentation in the updated version by emphasizing that Lemma 1 is an existing result rather than our own finding.
>
> > **Comment 3:** Can the results be extended to the adaptive adversary case?
>
> **Our response:** In the setting with full-information feedback, it has been shown that FTPL can achieve optimal regret even against adaptive adversaries (Hutter \& Poland, 2005). In the adversarial bandit literature, one common way to obtain regret guarantee against adaptive adversaries is via deriving high-probability regret guarantees against oblivious adversaries as in Lee et al. (2020) and Zimmert \& Lattimore (2022). Since the high-probability regret bound of FTPL has been established in Neu & Bartók (2016) for the standard bounded loss case, we tend to believe that it is indeed a promising future work to extend our results to adaptive adversaries in the heavy-tailed case.
>
> On the other hand, as pointed out by Arora et al. (2012), when adversaries can be adaptive, the intuitive meaning of "regret" no longer applies, as with respect to the original definition of regret, the authors showed that "while an algorithm minimizes regret, it certainly isn’t learning how to choose
> good actions to respond", and the proposed notion of "policy regret" is a more suitable metric. With this new definition, they show that "no bandit algorithm can guarantee a sublinear policy regret against an adaptive adversary with unbounded memory (i.e., the adversary can adapt to the entire history)". That is, it is impossible to have a no-regret algorithm under this new definition against an adaptive adversary.
>
> **References**
>
> - Dorn, Yuriy, Kornilov Nikita, Nikolay Kutuzov, Alexander Nazin, Eduard Gorbunov, and Alexander Gasnikov. "Implicitly normalized forecaster with clipping for linear and non-linear heavy-tailed multi-armed bandits." arXiv preprint arXiv:2305.06743 (2023).
>
> - Arora, Raman, Ofer Dekel, and Ambuj Tewari. "Online bandit learning against an adaptive adversary: from regret to policy regret." In Proceedings of the 29th International Conference on International Conference on Machine Learning, pp. 1747-1754. 2012.
>
> - Hutter, Marcus, and Jan Poland. "Adaptive Online Prediction by Following the Perturbed Leader." Journal of Machine Learning Research 6 (2005): 639-660.
>
> - Lee, Chung-Wei, Haipeng Luo, Chen-Yu Wei, and Mengxiao Zhang. "Bias no more: high-probability data-dependent regret bounds for adversarial bandits and mdps." Advances in neural information processing systems 33 (2020): 15522-15533.
>
> - Zimmert, Julian, and Tor Lattimore. "Return of the bias: Almost minimax optimal high probability bounds for adversarial linear bandits." In Conference on Learning Theory, pp. 3285-3312. PMLR, 2022.
>
> - Neu, Gergely, and Bartók Gábor. ”Importance weighting without importance weights: An efficient algorithm for combinatorial semi-bandits.” (2016).

---

> > ### Comment · Reviewer_PzqF · 2023-11-21
> >
> > Thanks for your reply. I have read your responses to my questions and I maintain my original score.

---

### Official Review · Reviewer_rKqQ · 2023-10-31

**Soundness:** 3 good
**Presentation:** 3 good
**Contribution:** 2 fair
**Rating:** 8
**Confidence:** 3

**Summary:**

The authors consider the adversarial bandit problems with heavy-tailed and possibly non-negative, bounded losses. The authors then point out the limitations of previously known work in this setting. The authors propose a novel FTPL solution scheme to tackle these challenges. This solution avoids the need for extra assumptions required by the state-of-the-art FTRL algorithm. Finally, the authors demonstrate the performance of the proposed algorithm by studying two examples: adversarial bandits with heavy-tailed losses and Huber contamination and  adversarial bandit with bounded loss  and additional Local Differential Privacy.

**Strengths:**

- The paper is very well-written and it is fairly easy to follow. The main idea is quite straightforward and the way it is presented (in comparison with previously known work) is very clear.

- The sketch of proof helps understanding the insights of the results and highlight the contributions (unfortunately, I did not have time to check the detailed proof in Appendix, but from the sketch, it seems sound enough).

- The two example applications are also well presented.

**Weaknesses:**

- A minor weakness is that the "breakthrough" comes from a known property (FTPL with Laplace perturbation) and its combination with Skipping (yet again, a previously known scheme) is rather obvious from the way it is currently presented. Personally, I believe that the authors do a good job in making such observations and tuning the algorithm's parameters (notably, L_t) which is often not that obvious. It might be better if the authors highlight a bit more on this aspect.
- The questions posed by the paper is interesting in a theoretical point of view. However, it might be better to provide some motivational/practical examples where it is essential to model the problem with negative, heavy-tailed losses like this instead of simply changing the loss models to fit more traditional non-negative loss framework.  The authors might argue that the Huber-contamination serves this purpose but Lemma 7 is valuable only if beta is significantly small, which undermines this argument.
- The proposed algorithm still requires knowledge of coefficients of Assumption 1 to deterministically tune the step-size, the skipping threshold and the L_t parameter. I am not sure how realistic this requirement can be.  As usual in bandit, it might be possible to relax this by using bounds of the involved coefficients instead of the true values. Do you think the results still hold?
- Another minor weakness of the algorithm is that the While-Loop that runs (in worst-case) with L_t rounds (that need be sufficiently large as pointed out) at each step. Is there possible to look for a better sampling scheme to check simultaneously many perturbed leaders at the same time?
- Despite being a theoretical paper, some simple experiments can help to clear further the comparison with SOTA (see questions below).

**Questions:**

- While I agree with Remark 10, Assumption 2 is purely for technical purpose (of the proof) and does not really hinder one to run the FTRL algorithm of Huang et al.2002 in your setting (even though their regret-bound is not guaranteed). It might be worth to run experiments to compare it with the proposed FTPL.

- Corollary 1 states the results specifically with the (epsilon-DP) mechanism. Does this mean it is not applicable to some situations where the privacy is not Laplace (but still guaranteeing heavy-tailed losses) ?

- Do you think it is possible to design a dynamic/adaptive skipping threshold r?

**Details Of Ethics Concerns:**

No ethical issues with this paper to signify.

---

> ### Author Response · Authors · 2023-11-17
> **Response to Reviewer rKqQ (Part 1)**
>
> We appreciate your time and your feedback.
>
> > **Comment 1:** The breakthrough comes from combining known properties/schemes. Our own "new" results should be highlighted more.
>
> **Our response:** We acknowledge that Lemma 1 (i.e., single-step stability with Laplace perturbation) is a known property and that we leverage the "skipping" scheme from the SOTA work of Huang et al. (2022). While this stability property is not new, as pointed out in our Remark 7, the original analysis has some gaps, and we believe that such a message is informative to the community towards building a better understanding of FTPL.
>
> We would also like to thank the reviewer for appreciating our work from other aspects, especially addressing the challenges in choosing $L_t$ and bounding the GR error; we could not follow the standard trick in the literature that works for non-negative losses only and thus propose a new workaround by leveraging the stability.
>
> > **Comment 2:** It might be better to provide some examples where it is essential to model the problem with negative, heavy-tailed losses like this instead of simply changing the loss models to fit the traditional non-negative loss framework.
>
> **Our response:** We are glad to know that the reviewer finds our work interesting from a theoretical point of view. Regarding your concern, it is true that one can simply shift all the losses if the effective range is bounded and known in advance, so that they fall into a non-negative range, and then one can apply the traditional framework for non-negative losses (see, e.g., Foster et al. (2020, Algorithm 3)).
>
> However, when the losses are heavy-tailed, there does not even exist such an effective bounded range that allows us to directly obtain non-negative losses via shifting. Moreover, even when the true losses are bounded and non-negative, the fundamental issue of extremely negative loss estimates could still exist in various online learning settings (e.g., due to the privacy mechanism in our private setting, as noted in Remark 12, or some other settings including loss shifting for the best-of-both-worlds (BOBW) guarantee and incorporating exploration bonus in Policy Optimization framework, as pointed out by Dann et al. (2023)).
>
> > **Comment 3:** Our algorithm and results require the knowledge of heavy tail parameters. Is it possible to relax this requirement by assuming the knowledge of the upper bounds on the parameters (and suffer regret guarantee depending on the quantities of parameter upper bounds)?
>
> **Our response:** The reviewer is correct. Our algorithm design and regret bound require the knowledge of heavy tail parameters $\sigma$ and $\alpha$, and in practice, these quantities may not be exactly known. If the underlying parameters $\sigma$ and $\alpha$ are unknown and we only know some upper (or lower) bounds $\sigma'$ and $\alpha'$, say, $\sigma'\geq \sigma > 0$ and $1<\alpha'\leq \alpha$, then our algorithm with $\sigma',\alpha'$ achieves a regret upper bound of $O(\sigma' K^{1-1/\alpha'}T^{1/\alpha'})$. That is, the optimality gap depends on how tight the upper (or lower) bounds $\sigma'$ and $\alpha'$ are.
>
> > **Comment 4:** The While-Loop could have a large runtime in the worst case (which is equal to $L_t$). Is possible to improve it?
>
> **Our response:** While the While-Loop could run for $L_t$ times in the worst case, we would like to mention that it only runs at most $K$ (recall that $K$ is the number of actions) times in expectation, regardless of the value of $L_t$ (Neu \& Bartók, 2016). Since the idea of GR is indeed based on Geometric distribution (i.e., repeating i.i.d. Bernoulli trials multiple times), we are not aware of any way to improve it by checking multiple new "perturbed leaders" simultaneously.
>
> However, the running time of the GR procedure can be improved slightly as follows: since we are interested in whether the played action $a_t$ is the minimizer of $\left(\tilde{z}\_i + \sum\_{t'=1}^{t-1}\hat{\ell}\_{t',i}\right)$, it is unnecessary to directly compute the minimizer of $ \left(\tilde{z}\_i + \sum_{t'=1}^{t-1}\widehat{\ell}\_{t',i}\right)$. Instead, one can first compute $\left(\tilde{z}\_{a_t} + \sum\_{t'=1}^{t-1}\widehat{\ell}\_{t',a_t}\right)$, i.e., the "perturbed (cumulative) loss" of action $a_t$. While one computes the perturbed losses of the other $K-1$ one by one, $a_t$ must not be the minimizer as soon as one finds any action that has a smaller perturbed loss (than $a_t$). Hence, one can move on to the next iteration without evaluating the cumulative losses of the remaining actions (if any).
>
> By doing so, although the worst-case running time remains unchanged, the empirical running time can be better due to the saved calculations in this procedure.

---

> ### Author Response · Authors · 2023-11-17
> **Response to Reviewer rKqQ (Part 2)**
>
> > **Comment 5:** Some simple experiments can help to clear further the comparison with SOTA.
>
> **Our response:** We agree that it is possible that this technical assumption may purely be an artifact from the analysis for the regret guarantee. Following the suggestion, we will conduct numerical studies and compare our algorithm with the FTRL-based algorithm of Huang et al. 2022. We may not be able to finish the experiments and share the results by the end of the rebuttal period (due to the short time window), we promise that we will include such results in the final version if our paper gets accepted.
>
> > **Comment 6:** Corollary 1 states the results specifically with the (epsilon-DP) mechanism. Does this mean it is not applicable to some situations where the privacy is not Laplace (but still guaranteeing heavy-tailed losses)?
>
> **Our response:** According to our definition of LDP bandits, it suffices to privatize every true loss by some $\epsilon$-DP mechanisms (which include but are not limited to the Laplace mechanism). Adopting any of these DP mechanisms ensures LDP. However, integrating different mechanisms into the design of the bandit algorithm could possibly lead to different regret analysis and/or guarantee. Given the known tail bounds of Laplace distribution, we chose the Laplace mechanism as the subroutine and presented its corresponding regret upper bound (which is implied by our general result for the heavy-tailed case). Of course, one can choose a different DP mechanism, e.g., the Randomized Response (Warner, 1965). While this mechanism could also result in negative losses for small $\epsilon$, our algorithm can still handle such negative losses and yield similar regret guarantees.
>
> > **Comment 7:** Is it possible to design an adaptive skipping threshold $r$?
>
> **Our response:** Note that the skipping threshold $r$ is in the form of $r=\sigma (T/K)^{1/\alpha}$, which depends on the heavy tail parameters $\alpha$ and $\sigma$. Since our algorithm already achieves the near-optimal regret when such heavy tail parameters are known, we believe that the reviewer is asking about the adaptive skipping threshold in the case of unknown heavy-tail parameters.
> In this case, since another reviewer (XopM) asked a similar question, we are adapting our answer there below.
>
> It is indeed not for free to adapt to the unknown heavy tail parameters, and the learner must suffer a (order-wise) worse regret than the known parameter case, due to some negative results from a very recent study by Genalti et al. (2023) (which was posted after the ICLR submission deadline). This paper shows that in stochastic bandits (which is, of course, a special case of adversarial bandits), "in general it is not possible to achieve the same order of performance while being adaptive to the unknown heavy tail parameters ($\sigma$ and $\alpha$)." Specifically, they show that if $\sigma$ or $\alpha$ is unknown, then in the worst case the learner must pay a worse regret (order-wise). Therefore, when heavy-tail parameters are unknown, it is not possible to still enjoy the same near-optimal regret as in the case of known heavy-tail parameters.
>
> **References**
>
> - Neu, Gergely, and Bartók Gábor. "Importance weighting without importance weights: An efficient algorithm for combinatorial semi-bandits." (2016).
>
> - Foster, Dylan J., Claudio Gentile, Mehryar Mohri, and Julian Zimmert. "Adapting to misspecification in contextual bandits." Advances in Neural Information Processing Systems 33 (2020): 11478-11489.
>
> - Dann, Christoph, Chen-Yu Wei, and Julian Zimmert. "Best of Both Worlds Policy Optimization." In Proceedings of the 40th International Conference on Machine Learning, 202:6968–7008. Proceedings of Machine Learning Research. PMLR, 23--29 Jul 2023.
>
> - Warner, Stanley L. "Randomized response: A survey technique for eliminating evasive answer bias." Journal of the American Statistical Association 60, no. 309 (1965): 63-69.
>
> - Genalti, Gianmarco, Lupo Marsigli, Nicola Gatti, and Alberto Maria Metelli. "Towards Fully Adaptive Regret Minimization in Heavy-Tailed Bandits." arXiv preprint arXiv:2310.02975 (2023).

---

> > ### Comment · Reviewer_rKqQ · 2023-11-22
> > **Responses to authors' feedbacks**
> >
> > The responses address all of my concerns. I would like to keep the original score.

---

### Official Review · Reviewer_J99q · 2023-10-31

**Soundness:** 3 good
**Presentation:** 2 fair
**Contribution:** 2 fair
**Rating:** 5
**Confidence:** 4

**Summary:**

The paper studies adversarial bandit problems with potentially heavy-tailed losses.
The authors propose a Follow-the-Perturbed-Leader (FTPL) based learning algorithm that achieves (nearly) optimal worst-case regret.
The authors further show that their algorithm works for adversarial bandits with heavy-tailed losses and Huber contamination and adversarial bandits in the private setting.

**Strengths:**

- The authors consider FTPL based alg. instead of FTRL based alg., which improves current results by $poly(\log T)$. Especially, the results of adversarial bandits with heavy-tailed losses and Huber contamination and adversarial bandits in the private setting are new and better than prior works.
- The paper is well writen. The proof in the appendix is well organized and mainly correct.

**Weaknesses:**

- Contrary to the author's statement, it is really trivial to use FTRL based algorithm to achieves (nearly) optimal worst-case regret for adversarial bandit problems with potentially heavy-tailed losses.
  First, using the similar skipping method as in the paper, the unbounded adversarial bandit problem can be reduced to the bounded case (the losses are bouned by $[-r,r]$, where $r = \sigma T^{1/\alpha} K^{-1/\alpha}$).
  Then, using the algorithm in [wei2018more] and scale the losses by $1/r$, by Theorem 4 in [wei2018more], we can immediately get regret upper bound $\mathcal{O}(K\ln T/\eta +\eta Q_{T, i*} + Kr (\ln T)^2 )$, where $Q_{T,i^*} = \sum_t (\ell_{t, i^*}-\sum_t \ell_{t, i^*}/T)^2\le \sum_t \ell_{t, i^*}^2$. Since here $|\ell_{t, i^*}|$ is upper bounded by $r$ due to the skippping, there is $\mathbb{E}[\sum_t \ell_{t, i^*}^2]\le \mathbb{E}[\sum_t |\ell_{t, i^*}|^\alpha r^{2-\alpha}] \le T \sigma^\alpha r^{2-\alpha} = \sigma^2 T^{2/\alpha} K^{1-2/\alpha} $.
  Thus, taking a suitable $\eta$ to balance the first and second terms results in regret $\mathcal{O}(\sigma T^{1/\alpha} K^{1-1/\alpha}(\ln T)^2 )$, which matches the results of this paper (up to log terms). (I guess such method can also get the regret guarantee Lemma 7 for adversarial bandits with huber contamination as in the paper)
- The design of GR count/GR maximum is confusing. According to the proof given by the authors, it is suffices to use the important weighting estimator (just set M_t = 1/w_{t, i}). In this case, GrErr goes to $0$ and FTPLReg can also be well bounded (proof of Lemma 5 still works). The authors do not clearly explain why it is important to use Geometric Resampling in their algorithm.

[wei2018more]: More Adaptive Algorithms for Adversarial Bandits

**Questions:**

- Why we need to use Geometric Resampling in the algorithm? Is it just for the proof of DP case?
- Is there any high level intuition why there exists poly(log K) in the regret? Is it possible to remove such terms?

**Details Of Ethics Concerns:**

Theory work. No ethics concerns.

---

> ### Author Response · Authors · 2023-11-17
> **Response to Reviewer J99q (Part 1)**
>
> We appreciate your time and your feedback.
>
> > **Comment 1:** The heavy-tailed bandits can be trivially solved by an algorithm from an algorithm by Wei \& Luo (2018, Theorem 4) (in particular, BROAD-OMD with particular choices of loss estimate, loss correction, and learning rate scheduling).
>
> **Our response:** Sorry for the confusion. We thank the reviewer for this detailed and sharp comment, which helps us improve the presentation of our work. First, we would like to clarify that we did not mean to claim that FTRL cannot handle negative losses. In fact, we were aware of and did acknowledge the capability of FTRL in handling negative losses with the log-barrier regularizer (as mentioned in Footnote \#5 in our original submission).
>
> However, we would like to thank the reviewer for offering the details about how to derive the near-optimal bound using existing results. To emphasize this point, in the updated version, we have highlighted this point in the main paper and also modified other texts on this point (e.g., the discussion on log-barrier in Related Work and Section 5) to avoid confusion. In addition, considering that this point may not be obvious to readers who are not familiar with the FTRL framework and the log-barrier regularizer, we have added the detailed derivations to Appendix F and acknowledged the reviewer.
>
> In the following, we explain that this fact does not weaken the significance of our contributions.
>
> While log-barrier can provably handle negative/heavy-tailed losses, they suffered an extra $\text{polylog}(T)$ factor compared to our results. Moreover, one key advantage of FTPL over FTRL is from the computational perspective: while FTRL requires solving a non-linear convex program over the probability simplex in each round (to determine the action), only a linear optimization problem is needed for FTPL, "this crucial difference between FTRL and FTPL makes the latter algorithm more attractive in practice" (Suggala \& Netrapalli, 2020a)). This advantage could be further amplified in non-convex online learning or dealing with adversarial MDPs (Suggala \& Netrapalli, 2020a; Dai et al., 2022).
>
> Because of such computational efficiency, the study of the under-explored FTPL framework in various online learning problems has been appreciated by the online learning community, even if the improvement in the theoretical guarantees of FTPL compared to FTRL is marginal or none (see, e.g., Dai et al. (2022), Wang \& Dong (2020), Suggala \& Netrapalli (2020b), and Honda et al. (2023)). Following this trend in the literature, we aim to investigate the use of FTPL in another important problem setup (namely, heavy-tailed bandits with important applications to the robust setting and the private setting).
>
> > **Comment 2:** It suffices to use the important weighting estimator. The use of Geometric Resampling (GR) is confusing.
>
> **Our response:** The reviewer is correct that if we simply construct the unbiased loss estimates in FTPL via the standard importance-weighted (IV) estimator as in FTRL, all the analyses and results still hold (and the GR Error is simply 0 since the estimates are unbiased). However, obtaining the closed-form solution of $w_t$ in FTPL is generally a key challenge, as pointed out by Abernethy et al. (2016), and we cannot apply the IV estimator without $w_t$. That is a main reason why we need GR as an alternative to construct the loss estimates as in Neu \& Bartók (2016) and Dai et al. (2022), and we have added this to Remark 4 in the revised version.
>
> > **Comment 3:** Any intuition on the extra $\sqrt{\ln K}$ factor in the upper bound? Is it possible to remove it?
>
> **Our response:** As mentioned in Remark 2, this $\sqrt{\ln K}$ factor is due to the perturbation distribution we choose (i.e., Laplace perturbation). In the standard bounded loss case, it was first conjectured by Kim and Ambuj (2019) that "the optimal perturbation, if it exists, will be of Fréchet-type." Later in the recent work by Honda et al. (2023), the authors show that the optimal regret with Fréchet-type perturbation is $O(\sqrt{KT})$. However, in the heavy-tailed setting we consider, the analysis relies on the specific single-step stability result (which is from and specific to Laplace perturbation), which enables us to handle negative losses and show the final regret upper bound. It is unclear as for how to adopt Fréchet-type perturbation in our framework to address the heavy-tailed losses.

---

> ### Author Response · Authors · 2023-11-17
> **Response to Reviewer J99q (Part 2)**
>
> **References**
>
> - Wei, Chen-Yu, and Haipeng Luo. "More adaptive algorithms for adversarial bandits." In Conference On Learning Theory, pp. 1263-1291. PMLR, 2018.
>
> - Abernethy, Jacob, Chansoo Lee, and Ambuj Tewari. "Perturbation techniques in online learning and optimization." Perturbations, Optimization, and Statistics 233 (2016).
>
> - Neu, Gergely, and Bartók Gábor. "Importance weighting without importance weights: An efficient algorithm for combinatorial semi-bandits." (2016).
>
> - Kim, Baekjin, and Ambuj Tewari. "On the optimality of perturbations in stochastic and adversarial multi-armed bandit problems." Advances in Neural Information Processing Systems 32 (2019).
>
> - Honda, Junya, Shinji Ito, and Taira Tsuchiya. "Follow-the-Perturbed-Leader Achieves Best-of-Both-Worlds for Bandit Problems." International Conference on Algorithmic Learning Theory. PMLR, 2023.
>
> - Dai, Yan, Haipeng Luo, and Liyu Chen. "Follow-the-Perturbed-Leader for Adversarial Markov Decision Processes with Bandit Feedback." In Advances in Neural Information Processing Systems, 2022.
>
> - Suggala, Arun, and Praneeth Netrapalli. "Follow the perturbed leader: Optimism and fast parallel algorithms for smooth minimax games." Advances in Neural Information Processing Systems 33 (2020a): 22316-22326.
>
> - Wang, Yuanhao, and Kefan Dong. "Refined analysis of fpl for adversarial markov decision processes." Theoretical Foundations of Reinforcement Learning Workshop, ICML, 2020.
>
> - Honda, Junya, Shinji Ito, and Taira Tsuchiya. "Follow-the-Perturbed-Leader Achieves Best-of-Both-Worlds for Bandit Problems." International Conference on Algorithmic Learning Theory. PMLR, 2023.
>
> - Suggala, Arun Sai, and Praneeth Netrapalli. "Online non-convex learning: Following the perturbed leader is optimal." In Algorithmic Learning Theory, pp. 845-861. PMLR, 2020b.

---

### Official Review · Reviewer_XopM · 2023-11-04

**Soundness:** 3 good
**Presentation:** 3 good
**Contribution:** 3 good
**Rating:** 6
**Confidence:** 4

**Summary:**

This paper proposes a FTPL-based algorithm for adversrial bandits with heavy-tailed losses. The proposed method achieves near-optimal regret and improves the regret of two applications: heavy-tailed adversarial badnits with huber contamization and adversarial bandits with bounded losses and LDP.

**Strengths:**

1. The paper is well-written and mostly clear.
2. The proposed method achieves near-optimal regret bound.

**Weaknesses:**

1. Notice that the FTRL algorithm is best-of-both-worlds (see Huang et al. 2022). Does the proposed algorithm can achieve optimal regret in the stochastic setting?
2. The "applications" in this paper are also bandit models. It would be better if the paper includes some empirical analysis of real world applications.

**Questions:**

Is it possible to design a near-optimal algorithm for bandits with heavy-tailed loss if we don't know the heavy-tail parameters $\sigma$ and $\alpha$?

---

> ### Author Response · Authors · 2023-11-17
> **Response to Reviewer XopM**
>
> We appreciate your time and your feedback.
>
> > **Comment 1:** Notice that the FTRL algorithm is best-of-both-worlds (see Huang et al. (2022)). Does the proposed algorithm can achieve optimal regret in the stochastic setting?
>
> **Our response:** Thank you for the question. It is still unclear whether the proposed algorithm can achieve optimal regret in the stochastic setting.
>
> On the one hand, note that in Huang et al. (2022), their best-of-both-worlds (BOBW) guarantee still requires the "truncated non-negativity" assumption. Specifically, this assumption serves two purposes: (1) stability and the regret guarantee in the adversarial setting (which is the focus of our work); (2) self-bounding property and regret guarantee in the stochastic setting (since it is not the focus of our work, it is not discussed in our paper; see Eq. (10) and Lemma A.5 in Huang et al. (2022) for the key steps). Since their analyses heavily rely on this assumption, it is unclear whether their algorithm can still achieve BOBW guarantees without such technical assumptions.
>
> On the other hand, to the best of our knowledge, the only existing result on the BOBW guarantee of FTPL is from a recent study by Honda et al. (2023) (but for the case of standard bounded losses), where they use a different type of perturbation (Fréchet-type perturbation) as mentioned in Remark 2 in our paper. However, their analyses rely heavily on the particular form of the Probability Density Function (PDF) for Fréchet-type perturbation. It remains unclear as for how to combine their work with ours to show BOBW guarantees under the FTPL framework in the heavy-tailed setup.
>
> > **Comment 2:** It would be better if the paper includes some empirical analysis of real world applications.
>
> **Our response:** Thank you for the suggestion. We will conduct simulations and include empirical results in future versions. Given the short time window, we may not be able to finish it and include the results by the end of the rebuttal period, but we promise that we will include them in the final version if it gets accepted.
>
> > **Comment 3:** Is it possible to design a near-optimal algorithm for bandits with heavy-tailed loss if we don't know the heavy-tail parameters?
>
> **Our response**: Thank you for the question. The answer is no, due to some negative results from a very recent study by Genalti et al. (2023, Theorems 2 and 3) (which was posted after the ICLR submission deadline). This paper shows that in stochastic bandits (which is, of course, a special case of adversarial bandits), "in general it is not possible to achieve the same order of performance while being adaptive to the unknown heavy tail parameters ($\sigma$ and $\alpha$)." Specifically, they show that if $\sigma$ or $\alpha$ is not known, then in the worst case the learner must pay a worse regret (order-wise). Therefore, it is not possible to still enjoy the near-optimal regret in the known parameter case when heavy tail parameters are unknown.
>
> **References**
>
> - Huang, Jiatai, Yan Dai, and Longbo Huang. "Adaptive best-of-both-worlds algorithm for heavy-tailed multi-armed bandits." In International Conference on Machine Learning, pp. 9173-9200. PMLR, 2022.
>
> - Honda, Junya, Shinji Ito, and Taira Tsuchiya. "Follow-the-Perturbed-Leader Achieves Best-of-Both-Worlds for Bandit Problems." International Conference on Algorithmic Learning Theory. PMLR, 2023.
>
> - Genalti, Gianmarco, Lupo Marsigli, Nicola Gatti, and Alberto Maria Metelli. "Towards Fully Adaptive Regret Minimization in Heavy-Tailed Bandits." arXiv preprint arXiv:2310.02975 (2023).

---

> > ### Comment · Reviewer_XopM · 2023-11-23
> >
> > Thank you for the reply. I will keep my score.

---

### Author Response · Authors · 2023-11-17
**General Response**

We want to thank all the reviewers for their time and their feedback. We have replied to each reviewer individually and provided an updated version of our paper by incorporating the reviewers' comments. We hope the reviewers find the response and the revision satisfactory, but we will be happy to clarify any further questions the reviewers may have.

The main changes we have made in the revised version (highlighted in blue) include the following:

- We have added the following explanations to Remark 4: (i) why the standard importance-weighted estimator is not applicable in FTPL and hence we need GR as an alternative for loss estimation; (ii) why we need to set the maximum GR run.

- We have explained that Lemma 1 (stability of FTPL) is a result of prior work rather than our own finding.

- We have clarified that log-barrier can be used to obtain near-optimal regret (yet up to a $(\ln T)^2$ factor) for FTRL/OMD in the heavy-tailed setup as pointed out by Reviewer J99q and added detailed derivations in Appendix F.

In the future versions, the changes we will make include the following:

- Providing simulation results and comparing the empirical performance of our algorithm and that of the FTRL-based algorithm from Huang et al. (2022).

**References**

- Huang, Jiatai, Yan Dai, and Longbo Huang. "Adaptive best-of-both-worlds algorithm for heavy-tailed multi-armed bandits." In International Conference on Machine Learning, pp. 9173-9200. PMLR, 2022.

---

### Note · Authors · 2024-04-14

**Comment:**

We would like to thank reviewers, AC, SAC, and PCs, for their efforts and time spent on the reviewing phase of our submission. We would also like to thank Yan Dai for pointing out a gap in the proof after the submission became public. It is unclear whether this gap can be fixed, with which the regret guarantee of our algorithm may not hold. Therefore, we are withdrawing our submission and leaving details in the message for clarification.

In this work, our main result is Theorem 1, i.e., a nearly minimax-optimal regret upper bound without the need for the truncated non-negativity assumption proposed in previous work. However, in the proof of Lemma 5, the first inequality is not true whenever $\widehat{\ell}_{t,i}$ is negative, which breaks our upper bound on the stability part. To address this gap, we need to arrive at step (b) therein. However, the current stability lemma (Lemma 4) may not be sufficient to show that.

The near-optimal regrets of our algorithm in the privacy setting and robust setting may not hold either as they are implications of the main result above. However, since these two regrets are obtained based on a reduction argument, if there’s a near-optimal algorithm in the heavy-tailed setup, one may still be able to establish these implications. Moreover, the lower bounds (Theorem 2 in Appendix B.4.2 and Theorem 3 in Appendix C.2) we provided are not affected by the failure of the upper bounds.



Regards,

Authors of Submission 7706

(Duo Cheng, Xingyu Zhou, Bo Ji)

**Withdrawal Confirmation:**

I have read and agree with the venue's withdrawal policy on behalf of myself and my co-authors.

---

### Meta-Review · Area_Chair_LQgV · 2023-12-19

**Metareview:**

The paper considers the problem of adversarial bandits where losses are not assumed to be bounded but rather are considered to be distributions with heavy tails. This setting has been studied before via FTRL algorithms and the main challenge comes from handling large negative losses. Previous SOTA work (Huang et al) requires a truncated negative type assumption which restricts the class of losses. This paper proposes a neat FTPL based algorithm with a simple and known but key insight on the stability which leads to the result without any assumptions but heavy tails. Further applications to bandits with Huber contamination and LDP with bandits are provided.

The paper is clearly written and the reviewers acknowledged and appreciated the contributions. FTPL style algorithms are also more efficient which is another plus. Overall the paper makes a good contribution to this line of work and I recommend acceptance.

IMPORTANT NOTE TO AUTHORS -- I do not agree are still crediting the known OMD result with log barrier. In particular their abstract still says "Notably, our method achieves (near-)optimal worst-case regret, eliminating the need for an undesired assumption inherent in the Follow-the-Regularized-Leader (FTRL) based approach proposed in the prior work." This gives the impression that no previous work solved this problem at all which is False. I would suggest a more clear statement regarding this. You must also add a discussion of this in Section 1 and not later in related work. This result is required to situate your work clearly.

**Justification For Why Not Higher Score:**

The existence of the OMD result weakens the novel claim by the paper which is now based on the efficiency gain via FTPL. Furthermore the paper in my opinion is yet not doing a good enough job of highlighting this result. See my meta review. Overall thereby I and the reviewers think that the paper holds merit for accepatance but not enough for a spotlight.

**Justification For Why Not Lower Score:**

The reviewers found the results to be meaningful interesting and useful and consistently gave the paper an above accept rating.

---

### Decision · Program_Chairs · 2024-01-16

Accept (poster)